# COMPRESSION OF VISION TRANSFORMER BY REDUCTION OF KERNEL COMPLEXITY

## ABSTRACT

Self-attention and transformer architectures have become foundational components in modern deep learning. Recent efforts have integrated transformer blocks into compact neural architectures for computer vision, giving rise to various efficient vision transformers. In this work, we introduce Transformer with Kernel Complexity Reduction, or KCR-Transformer, a compact transformer block equipped with differentiable channel selection, guided by a novel and sharp theoretical generalization bound. To reduce the substantial computational cost of the MLP layers, the KCR-Transformer performs channel selection on the outputs of its self-attention layer. Furthermore, we provide a rigorous theoretical analysis establishing a tight generalization bound for networks equipped with KCR-Transformer blocks. Leveraging such strong theoretical results, the channel pruning by KCR-Transformer is conducted in a generalization-aware manner, ensuring that the resulting network retains a provably small generalization error. Our KCR-Transformer is compatible with many popular and compact transformer networks, such as ViT and Swin, and it reduces the FLOPs of the vision transformers while maintaining or even improving the prediction accuracy. In the experiments, we replace all the transformer blocks in the vision transformers with KCR-Transformer blocks, leading to KCR-Transformer networks with different backbones. The resulting KCR-Transformers achieve superior performance on various computer vision tasks, achieving even better performance than the original models with even less FLOPs and parameters. The code of the KCR-Transformer is available at https://anonymous.4open.science/status/KCR-Transformer.

## 1 INTRODUCTION

Vision transformers have demonstrated promising performance on a variety of computer vision tasks (Yuan et al., 2021; Dosovitskiy et al., 2021; Liu et al., 2021a; Zhu et al., 2021; Liang et al., 2021) and multi-modal learning tasks (Liu et al., 2023a; Singh et al., 2022). However, the superior performance of vision transformers comes at the cost of substantial computational overhead (Dosovitskiy et al., 2021; Touvron et al., 2021). To reduce the computational costs of the vision transformers, various model compression methods have been developed, including knowledge distillation (Zhao et al., 2023; Yang et al., 2023), quantization (Li et al., 2022a; Lin et al., 2022; Li et al., 2023; Liu et al., 2021b), neural architecture search (Gong et al., 2022; Su et al., 2022), and pruning (Chen et al., 2021b; Chavan et al., 2022; Zheng et al., 2022; Yu et al., 2022b;a; Rao et al., 2021; Kong et al., 2022; Wang et al., 2022; Bolya et al., 2023; Bonnaerens & Dambre, 2023; Kim et al., 2024). The pruning methods, which typically involve a pruning stage to remove redundant parameters and a fine-tuning stage to recover performance, have been shown to be particularly effective due to the substantial parameter redundancy in ViT models (Chen et al., 2021b; Chavan et al., 2022; Zheng et al., 2022; Yu et al., 2022b;a; Rao et al., 2021; Kong et al., 2022; Wang et al., 2022; Bolya et al., 2023; Bonnaerens & Dambre, 2023; Kim et al., 2024). Despite substantial progress, the conventional model compression methods primarily focus on identifying optimal compression strategies through direct performance–efficiency trade-offs, guided by empirical heuristics rather than principled theoretical foundations.

More recently, inspired by advances in the theoretical understanding of deep neural networks (DNNs) through the lens of kernel methods, such as the Neural Tangent Kernel (NTK) (Jacot et al.,

2018), the kernel-based compression method has emerged as a principled alternative. The kernel-based methods aim to preserve the generalization capability of the compressed model by ensuring that the compressed model retains the training dynamics, convergence behavior, and inductive biases of the original network through spectral alignment of the NTK (Wei et al., 2023; Chen et al., 2021c; Mok et al., 2022; Wang et al., 2023; Rachwan et al., 2022). For instance, NTK-SAP (Wang et al., 2023) and Early-Lottery (Rachwan et al., 2022) leverage spectral preservation of the NTK during pruning to maintain the eigenspectrum, thereby preserving the generalization characteristics of the NTK of the original DNNs. Despite recent progress, a substantial gap persists between theory and practice in enhancing the generalization capability of compressed DNNs through NTK-based kernel learning. Existing theoretical frameworks, particularly those grounded in NTK analyses, are predominantly limited to over-parameterized DNNs (Cao & Gu, 2019; Arora et al., 2019; Ghorbani et al., 2021), rendering them unsuited for modern architectures such as vision transformers with finite-width and diverse network architectures. Moreover, the linearized nature of the NTK regime inherently fails to model the dynamically evolving kernel characteristic of realistic training dynamics, thereby limiting its applicability to compressed models utilizing large-scale training for compelling performance on real-world tasks (Nichani et al., 2022; Damian et al., 2022; Takakura & Suzuki, 2024). To address this challenge, we first provide a theoretical analysis that establishes tight generalization upper and lower bounds in Theorem 3.1. Both an upper and lower bound of the expected loss, referred to as KCR upper and lower bounds, are established based on the training loss and the kernel complexity (KC) of the kernel gram matrix computed over the training data. The tightness of the KCR upper and lower bounds is studied in Section 4.6. *In contrast to the current NTK-based methods with their feature learning capability limited by the linear region of NTK, the KC measures the complexity of the dynamically evolving kernel formed by the DNN during the training process, accommodating the rich feature learning capability of DNNs.* Since the training loss is usually optimized to a small value by training the DNN, the KCR upper and lower bounds in Theorem 3.1 can be tight and close to the expected loss if the KC is small. However, the computation of the KC involves the costly computation of the eigenvalues of a potentially large-scale gram matrix. To mitigate this issue, we introduce an approximate TNN through the efficient Nyström method (Kumar et al., 2012), and the KC is reduced by the reduction of the approximate TNN. The approximate TNN is computed efficiently as a regularization term to the regular cross-entropy training loss. Since the approximate TNN is separable, it can be optimized by the standard SGD-based optimization algorithms. Based on the reduction of the KC, we propose a novel vision transformer termed the Transformer with Kernel Complexity Reduction, or KCR-Transformer. The training of the KCR-Transformer involves a search stage and a retrain stage. The channel selection in the input/output features of the MLP in the transformer block of the KCR-Transformer is performed in the search stage to obtain a compressed network architecture with reduced computation costs. To guarantee the generalization capability of the compressed model, the compressed network is retrained with the approximate TNN as a regularization term to reduce the KC, leading to enhanced prediction accuracy. It is verified through extensive experiments that the reduction of the approximate TNN effectively reduces the KC.

**Contributions.** The contributions of this paper are presented as follows.

First, we present a compact transformer block termed Transformer with Kernel Complexity Reduction, or KCR-Transformer. By selecting the channels in the input of the MLP layers via channel pruning, KCR-Transformer blocks effectively reduce the computational costs of the vision transformer. KCR-Transformer blocks can be used to replace all the transformer blocks in many popular vision transformers, rendering compact KCR-Transformer networks with comparable or even better performance. The effectiveness of KCR-Transformer is evidenced by replacing all the transformer blocks with KCR-Transformer blocks in popular vision transformers, such as ViT (Dosovitskiy et al., 2021), and Swin (Liu et al., 2021a), rendering compact models with competitive performance. Experimental results show that KCR-Transformer not only reduces the parameter size and FLOPs but also outperforms the original models on tasks including image classification, object detection, instance segmentation, and visual question answering.

Second, we provide a theoretical analysis showing tight generalization upper and lower bounds for the KCR-Transformer network. With such strong theoretical results in Theorem 3.1, the channel pruning by KCR-Transformer is performed in a *generalization-aware* manner. That is, the channel pruning of KCR provably keeps a small generalization error bound for the DNN with KCR-Transformer blocks, effectively guaranteeing the generalization capability of the DNN after channel

pruning. This goal is achieved through the reduction of the KC of the DNN with KCR-Transformer blocks. Since the KC involves the computation of the eigenvalues of a potentially large gram matrix, we introduce an approximate truncated nuclear norm (TNN) through the Nyström method (Kumar et al., 2012), which is computed efficiently as a regularization term to the regular cross-entropy training loss and separable, so that it can be optimized by the standard SGD-based optimization algorithms. The reduction of the approximate TNN effectively reduces the KC, leading to the superior prediction accuracy of the compressed vision transformers by KCR-Transformer. Furthermore, unlike existing NTK-based compression methods constrained by NTK's linear regime, our KC captures the complexity of the evolving DNN kernel, enabling its rich feature learning capacity.

We note that training KCR-Transformer networks with the KCR regularization is efficient and stable with respect to the regularization weight, as evidenced in Section D.5 of the appendix. This paper is organized as follows. The related works in efficient vision transformers and compression of vision transformers are discussed in Section 2. The formulation of KCR-Transformer with our theoretical results is detailed in Section 3. The effectiveness of KCR-Transformer is demonstrated in Section 4 for image classification, dense prediction tasks, and multi-modal learning tasks. Throughout this paper, we use $a \lesssim b$ to denote $a \leq Cb$ if there exists such a positive constant $C$, and $a = \Theta(b)$ indicates that $a \lesssim b$ and $b \lesssim a$. $[n]$ denotes all the natural numbers between $1$ and $n$ inclusively.

## 2 RELATED WORKS

### 2.1 EFFICIENT VISION TRANSFORMERS AND COMPRESSION OF VISION TRANSFORMERS

To mitigate the issue of substantial computational overhead of the vision transformers, sparse attention mechanisms have been introduced to reduce computational demands (Zhu et al., 2021; Yuan et al., 2021; Papa et al., 2024), while other efforts integrate convolutional operations into the transformer architecture (Cai et al., 2023; Mehta & Rastegari, 2022; Yuan et al., 2021; Bravo-Ortiz et al., 2024). Additional gains in efficiency have been realized through Neural Architecture Search (Chen et al., 2021a; Gong et al., 2022; Wei et al., 2024) and Knowledge Distillation (Graham et al., 2021; Radosavovic et al., 2020; Gong et al., 2022; Yang et al., 2024), which aim to maintain accuracy with reduced computational resources. To further compress vision transformers, pruning techniques have been extensively explored. Channel pruning aims to eliminate redundant attention heads and channels (Chen et al., 2021b; Chavan et al., 2022; Zheng et al., 2022; Xu et al., 2024; Ahmed et al., 2025). Block pruning reduces the depth and width of models by removing entire transformer blocks (Yu et al., 2022b;a; Liu et al., 2024a). Token pruning techniques improve efficiency by adaptively discarding, merging, or filtering less informative tokens (Rao et al., 2021; Kong et al., 2022; Bolya et al., 2023; Wang et al., 2022; Liu et al., 2024b; Mao et al., 2025).

### 2.2 RELATED WORKS ABOUT KERNEL METHODS FOR DEEP NEURAL NETWORKS (DNNS)

Kernel methods have offered a principled view for analyzing the training dynamics, generalization properties, and architectural components of DNNs. One of the most prominent lines of work centers on the neural tangent kernel (NTK) (Jacot et al., 2018). Subsequent studies have extended NTK theory to better capture realistic scenarios, including finite-width settings (Seleznova & Kutyniok, 2022), deep narrow networks (Lee et al., 2022), and the empirical evolution of the NTK during training (Fort et al., 2020). Following these, researchers have also studied the limitations of purely kernel-based theories (Woodworth et al., 2020; Barrett & Dherin, 2021). Recent works have examined kernel-based interpretations of feature learning and generalization, revealing how hierarchical or implicit kernel structures emerge within deep models (Montavon et al., 2011; Belkin et al., 2018; Xiao et al., 2020; Canatar & Pehlevan, 2022; Deng et al., 2022). Building on these theoretical foundations, recent efforts propose reproducing kernel Hilbert space (RKHS) representations and operator-theoretic formulations as a basis for deep learning (Hashimoto et al., 2023), and develop hierarchical kernels tailored for representation learning (Huang et al., 2023). Beyond theoretical analysis, the study of kernels has also inspired reinterpretations and enhancements of transformer architectures. Several studies formulate self-attention as a kernel operation (Song et al., 2021; Chen et al., 2023). Others leverage spectral or integral transforms grounded in kernel theory (Nguyen et al., 2022; 2023). Positional encoding has also benefited from this perspective, with kernelized relative embeddings proposed for improved sequence extrapolation (Chi et al., 2022). Efficient attention variant, Performer (Choromanski et al., 2021), exploits kernel approximations to achieve linear complexity while maintaining expressiveness. Additionally, kernel-based models have been used to improve calibration in transformers via sparse Gaussian processes (Chen & Li, 2023).

**Kernel-Based Model Compression Methods.** Building upon these insights, kernel-based methods, especially those centered on the NTK, provide a complementary theoretical framework for analyzing and guiding model compression. The NTK-Comp framework (Gu et al., 2022) investigates pruning in wide multilayer perceptrons under Gaussian input assumptions and introduces quantization techniques that preserve the NTK spectrum within linear layers. MLP-Fusion (Wei et al., 2023) advances LLM compression by clustering neurons to jointly approximate functional outputs and NTK similarity. NTK-based metrics have also enabled training-free architecture search (Chen et al., 2021c), and facilitated early-stage performance prediction in neural architecture search (Mok et al., 2022), though their predictive power may diminish in regimes dominated by highly non-linear dynamics. In addition, methods, such as NTK-SAP (Wang et al., 2023) and Early-Lottery (Rachwan et al., 2022), further highlight the importance of preserving NTK spectral properties during pruning, emphasizing spectral alignment as critical for maintaining stable training dynamics. Nonetheless, the core limitation of existing NTK-based compression methods lies in their dependence on static or "lazy" training regimes, limiting their applicability to models with dynamically evolving representations.

## 3 FORMULATION

In this section, we introduce the KCR-Transformer, a compact transformer block designed to reduce the computational overhead of vision transformers through differentiable channel pruning in the MLP layers. To guide this pruning in a theoretically grounded manner, we present a novel generalization bound based on the kernel complexity (KC) of the network, and introduce the training algorithm of the network with KCR-Transformer for minimizing the upper bound.

### 3.1 CHANNEL SELECTION FOR ATTENTION OUTPUTS

The vision transformer blocks usually apply a series of MLP layers to the output of the multi-head self-attention, which incurs substantial computation costs. To improve the efficiency of the vision transformer block, we propose pruning the channels in the attention outputs, thereby reducing the computation cost of the MLP layers, leading to the compact KCR-Transformer. To this end, we maintain a decision mask $g_i \in \{0, 1\}^D$, where $g_i = 1$ indicates that the $i$-th channel is selected, and $0$ otherwise. Thus, the informative channels can be selected by multiplying $g$ by each row of the attention output. To optimize the binary decision mask with gradient descent, we replace $g$ with Gumbel Softmax weights in the continuous domain, which is computed by $\widehat{g}_i = \sigma\left(\frac{\alpha_i + \epsilon_i^{(1)} - \epsilon_i^{(2)}}{\tau}\right)$, where $\epsilon_i^{(1)}$ and $\epsilon_i^{(2)}$ are Gumbel noise. $\tau$ is the temperature. $\alpha \in \mathbb{R}^D$ is the sampling parameter. We define $\alpha$ as the architecture parameters that can be optimized by gradient descent during the differentiable search process. By gradually decreasing the temperature $\tau$ in the search process, $\alpha_i$ will be optimized such that $g_i$ will approach $1$ or $0$. Note that since the MLP layers in vision transformers have the same input and output dimensions, we multiply the decision mask $g$ with both the input and output features of the MLP layers. After the search is finished, we apply the gather operation on the attention outputs from the selected channels. The dimension of the input and output features of the MLP layers is then changed to $\tilde{D} = \sum_{i=1}^{D} g_i$.

### 3.2 GENERALIZATION BOUND BY APPROXIMATE KERNEL COMPLEXITY LOSS AND ITS OPTIMIZATION

Given the training data $\{\mathbf{x}_i, y_i\}_{i=1}^n$ where $\mathbf{x}_i$ is the $i$-th input training feature, $\mathbf{y}_i \in \mathbb{R}^C$ is the corresponding one-hot vector as the class label of $\mathbf{x}_i$ and $C$ is the number of classes. We denote the label matrix as $\mathbf{Y} \in \mathbb{R}^{n \times C}$ where the $i$-th row is $\mathbf{y}_i$ for all $i \in [n]$. Let $\mathbf{F} \in \mathbb{R}^{n \times d}$ be the features extracted on the entire training data set, where $g(\cdot) \in \mathbb{R}^d$ denotes the mapping function of a DNN, such as ViT (Dosovitskiy et al., 2021), and $d$ is the output dimension of the DNN before the final softmax layer for classification. We remark that almost all the DNNs use a linear layer to generate the output of the network for discriminative learning tasks, so that the mapping function of a DNN can be formulated as $g(\cdot) = g(\mathcal{W}, \cdot) = F(\mathcal{W}_2, \cdot)\mathcal{W}_1$, where $\mathcal{W}_1 \in \mathbb{R}^{d \times C}$ contains the weights in the final linear layer of the DNN where $d$ is the hidden dimension, $F(\mathcal{W}_2, \cdot) \in \mathbb{R}^d$ represents the feature extraction backbone of the network before the final linear layer, and $\mathcal{W}_2$ are the weights of the backbone $F$, with $\mathcal{W} = \{\mathcal{W}_1, \mathcal{W}_2\}$. The DNN is denoted as $\mathrm{NN}_{\mathcal{W}}(\cdot)$. It is noted that such a formulation does not impose any limitation on the feature backbone $F$ so as to admit a broad class of DNNs with various architectures for real-world vision discriminative tasks. We define a positive definite kernel for the DNN as $K(\mathbf{x}, \mathbf{x}') = F(\mathbf{x})^\top F(\mathbf{x}')$, where $\mathcal{X}$ is the input domain of the DNN

and $F(\mathcal{W}, \cdot)$ is abbreviated into $F(\cdot)$. Compared to existing NTK-based methods (Wang et al., 2023; Wei et al., 2023; Gu et al., 2022) where a static NTK is used as the kernel, our kernel $K$ is dynamic with the learned feature backbone $F$ during the training process of the DNN. Let $\mathbf{F}_i = F(\mathcal{W}_2, \mathbf{x}_i)$ be the learned representation for the $i$-th training data, and $\mathbf{F} \in \mathbb{R}^{n \times d}$ is the matrix of all the learned representations over the training data. Then the gram matrix $\mathbf{K} \in \mathbb{R}^{n \times n}$ of the kernel $K$ over the training data is calculated by $\mathbf{K} = \mathbf{F}\mathbf{F}^\top \in \mathbb{R}^{n \times n}$, and the eigenvalues of $\mathbf{K}_n \coloneqq \mathbf{K}/n$ are $\widehat{\lambda}_1 \geq \widehat{\lambda}_2 \ldots \geq \widehat{\lambda}_{r_0} \geq \widehat{\lambda}_{r_0+1} = \ldots = \widehat{\lambda}_n = 0$ with $r_0 = \min\{n, d\}$, since $\mathrm{rank}(\mathbf{K}) \leq r_0$.

Suppose the input feature $\mathbf{x}$ and its class label vector $\mathbf{y}$ follow an unknown joint distribution $P$, then the expected risk of the DNN is defined as $L_{\mathcal{D}}(\mathrm{NN}_{\mathcal{W}}) = \mathbb{E}_{(\mathbf{x},\mathbf{y}) \in P}\left[\|\mathrm{NN}_{\mathcal{W}}(\mathbf{x}) - \mathbf{y}\|_2^2\right]$, which also represents the generalization error of the DNN. The following theorem, Theorem 3.1, based on the local complexity of the function class of the DNN feature extraction backbones and rooted in the well-established local Rademacher complexity literature (Bartlett et al., 2005; Koltchinskii, 2006; Mendelson, 2002), gives sharp upper and lower bounds for the generalization error of the DNN. Theorem 3.1 uses the kernel complexity (KC) of the dynamic kernel $K$ over the training data as

$$\mathrm{KC}(\mathbf{K}) \coloneqq \min_{h \in [0, r_0]} \left( \frac{h}{n} + \sqrt{\frac{1}{n} \sum_{i=h+1}^{r_0} \widehat{\lambda}_i} \right).$$

**Theorem 3.1.** Let $K$ be the dynamic kernel after a particular optimization epoch by GD or SGD. Then for every $x > 0$, with probability at least $1 - \exp(-x)$, we have

$$\underbrace{\mathbb{E}_{P_n}\left[\|\mathrm{NN}_{\mathcal{W}}(\mathbf{x}) - \mathbf{y}\|_2^2\right] - \mathrm{KC}(\mathbf{K}) - \frac{x}{n}}_{\text{KCR Lower Bound}} \lesssim L_{\mathcal{D}}(\mathrm{NN}_{\mathcal{W}}) \lesssim \underbrace{\mathbb{E}_{P_n}\left[\|\mathrm{NN}_{\mathcal{W}}(\mathbf{x}) - \mathbf{y}\|_2^2\right] + \mathrm{KC}(\mathbf{K}) + \frac{x}{n}}_{\text{KCR Upper Bound}},$$

(1)

where $\mathbb{E}_{P_n}\left[\|\mathrm{NN}_{\mathcal{W}}(\mathbf{x}) - \mathbf{y}\|_2^2\right] = 1/n \cdot \sum_{i=1}^n \|\mathrm{NN}_{\mathcal{W}}(\mathbf{x}_i) - \mathbf{y}_i\|_2^2$ is the empirical loss on the training data.

The proof of this theorem is deferred to Section A of the appendix. The empirical loss $\mathbb{E}_{P_n}\left[\|\mathrm{NN}_{\mathcal{W}}(\mathbf{x}) - \mathbf{y}\|_2^2\right]$ is usually optimized to a small value by training the network $\mathrm{NN}_{\mathcal{W}}$, so it can be observed from (1) that the KCR upper and lower bounds for $L_{\mathcal{D}}(\mathrm{NN}_{\mathcal{W}})$ can be tight and close to $L_{\mathcal{D}}(\mathrm{NN}_{\mathcal{W}})$ if $\mathrm{KC}(\mathbf{K})$ is small, guaranteeing the generalization capability of the DNN. To reduce the KC, we introduce the truncated nuclear norm (TNN) of $\mathbf{K}$ denoted as $\|\mathbf{K}\|_r \coloneqq \sum_{i=r+1}^{r_0} \widehat{\lambda}_i$ where $r \in [0 \colon r_0]$, with $r_0 = \gamma_0 \min\{n, d\}$. $\gamma_0$ is to be chosen by cross-validation. It can be verified that a smaller TNN $\|\mathbf{K}\|_r$ leads to a smaller KC, $\mathrm{KC}(\mathbf{K})$. Since the computation of the TNN involves the computation of the eigenvalues of the potentially large-scale gram matrix $\mathbf{K}$ over large-scale training data, we then describe below how to efficiently and effectively approximate the TNN. We first compute the approximate top-$r_0$ eigenvectors of $\mathbf{K}_n$, $\tilde{\mathbf{U}}^{(r_0)}$, by the Nyström method (Kumar et al., 2012). Here $\mathbf{A}^{(r)}$ denotes a submatrix of $\mathbf{A}$ formed by its top $r$ columns.

**Efficient Computation of the Top-$r_0$ Eigenvectors of K.** To approximate the top-$r_0$ eigenvectors, $\mathbf{U}^{(r_0)} \in \mathbb{R}^{n \times r_0}$, of the gram matrix $\mathbf{K}$ using the Nyström method (Kumar et al., 2012), we first sample $m$ landmark points from the training set, indexed by $\mathcal{I} \subset [n]$ with $|\mathcal{I}| = m \ll n$. Let $\mathbf{F}_{\mathcal{I}} \in \mathbb{R}^{m \times d}$ be the features corresponding to the landmark set. We define $\mathbf{C} = \mathbf{F}\mathbf{F}_{\mathcal{I}}^\top \in \mathbb{R}^{n \times m}$ as the cross-covariance matrix and $\mathcal{W} = \mathbf{F}_{\mathcal{I}}\mathbf{F}_{\mathcal{I}}^\top \in \mathbb{R}^{m \times m}$ as the gram matrix on the landmarks. Next, we compute the top-$r_0$ eigen-decomposition of $\mathcal{W}$ as $\mathcal{W} = \mathbf{Q}\mathbf{\Lambda}\mathbf{Q}^\top$, where $\mathbf{Q} \in \mathbb{R}^{m \times r_0}$ contains the top-$r_0$ eigenvectors and $\mathbf{\Lambda} \in \mathbb{R}^{r_0 \times r_0}$ is the diagonal matrix of corresponding eigenvalues. The Nyström approximation of $\mathbf{K}$ is then given by $\tilde{\mathbf{K}} = \mathbf{C}\mathcal{W}^\dagger\mathbf{C}^\top$, and the approximate top-$r_0$ eigenvectors are computed as $\tilde{\mathbf{U}}^{(r_0)} = \mathbf{C}\mathbf{Q}\mathbf{\Lambda}^{-1/2} \in \mathbb{R}^{n \times r_0}$, which serves as an efficient approximation to $\mathbf{U}^{(r_0)}$ with significantly reduced computational cost.

We let $\mathbf{U}_r = \tilde{\mathbf{U}}^{(r)}$, then the sum of the top-$r$ eigenvalues of $\mathbf{K}_n$ is approximated by $\mathrm{tr}\left(\mathbf{U}_r^\top \mathbf{K}_n \mathbf{U}_r\right)$. Since $\mathrm{tr}(\mathbf{K}_n) = \sum_{i=1}^n K(\mathbf{x}_i, \mathbf{x}_i) = \sum_{i=1}^n \widehat{\lambda}_i$, and $\mathrm{tr}\left(\mathbf{U}_r^\top \mathbf{K}_n \mathbf{U}_r\right) = \sum_{i=1}^n \left(\sum_{s=1}^r \sum_{k=1}^n [\mathbf{U}_r]_{si}^\top [\mathbf{K}_n]_{ik} [\mathbf{U}_r]_{ks}\right)$, we can approximate the $\|\mathbf{K}\|_r$ by $\overline{\|\mathbf{K}\|_r} = \mathrm{tr}(\mathbf{K}_n) - \mathrm{tr}\left(\mathbf{U}_r^\top \mathbf{K}_n \mathbf{U}_r\right)$ which is separable. In particular, $\overline{\|\mathbf{K}\|_r} = \sum_{i=1}^n K(\mathbf{x}_i, \mathbf{x}_i) -$

$\sum_{i=1}^{n} \left( \sum_{s=1}^{r} \sum_{k=1}^{n} [\mathbf{U}_r]_{si}^{\top} [\mathbf{K}_n]_{ik} [\mathbf{U}_r]_{ks} \right)$. We remark that $\overline{\|\mathbf{K}\|}_r$ is ready to be optimized by standard SGD algorithms because it is separable and expressed as the summation of losses on individual training data points. The training loss of a neural network with KCR-Transformer blocks has $\text{KCR}(\mathcal{W}) = \overline{\|\mathbf{K}\|}_r$, the approximate TNN, as a regularization term. The following functions are needed for minibatch-based training with SGD, with the subscript $j$ indicating the corresponding loss on the $j$-th batch $\mathcal{B}_j$:

$$\text{KCR}_j(\mathcal{W}) = \frac{1}{|\mathcal{B}_j|} \sum_{i=1}^{|\mathcal{B}_j|} \left( K(\mathbf{x}_i, \mathbf{x}_i) - \sum_{s=1}^{r} \sum_{k=1}^{n} [\mathbf{U}_r]_{si}^{\top} [\mathbf{K}_n]_{ik} [\mathbf{U}_r]_{ks} \right), \text{CE}_j^{(t)}(\mathcal{W}) = \frac{1}{|\mathcal{B}_j|} \sum_{i=1}^{|\mathcal{B}_j|} H(X_i(\mathcal{W}), Y_i),$$

$$\mathcal{L}_{\text{train},j}^{(t)}(\mathcal{W}) = \text{CE}_j^{(t)}(\mathcal{W}) + \eta \text{KCR}_j(\mathcal{W}). \tag{2}$$

Here $\text{CE}_j^{(t)}$ is the cross-entropy loss on batch $\mathcal{B}_j$ at epoch $t$. $H(,)$ is the cross-entropy function. $\eta$ is the balance factor.

### 3.3 Search and Retraining Processes

To obtain a compact vision transformer network with KCR-Transformer, or the KCR-Transformer network, we optimize both the accuracy of the network and the inference cost (FLOPs) of the network. We follow the standard techniques in neural architecture search (Tan et al., 2019), where the attention output channels are pruned in the search process, and the pruned network is retrained in the retraining process. The search process involves channel selection by Gumbel-Softmax and entropy minimization for architecture search. We optimize the FLOPs of the operations whose computation cost is decided by the channel selection for attention outputs described in Section 3.1. We estimate the FLOPs of the MLP after the channel selection on the attention outputs following the KCR-Transformer. The FLOPs related to a single Transformer block is $\text{cost}_j = l_j \cdot \left( 2\tilde{D}^2 + \tilde{D} \right)$, where $j$ indexes the KCR-Transformer block. $2\tilde{D}^2 + \tilde{D}$ is the FLOPs of a layer of the MLP after the channel selection on the attention outputs, and $l_j$ denotes the number of layers in the MLP of the $j$-th KCR-Transformer block. The inference cost objective of the network architecture is computed by $\text{cost} = \sum_{j=1}^{M} \text{cost}_j$, where $M$ is the number of transformer blocks. The overall loss function for search on each batch $\mathcal{B}_j$ at epoch $t$ is formulated by $\mathcal{L}_{\text{search},j}^{(t)}(\mathcal{W}, \alpha) = \mathcal{L}_{\text{train},j}^{(t)}(\mathcal{W}) + \lambda \cdot \log \text{cost}(\alpha)$, where $\alpha$ is the architecture parameter. $\lambda$ controls the magnitude of the cost term. In the search phase, the search loss is optimized to perform the two types of channel selection for all the KCR-Transformer blocks. After the search process, we use the selected channels for the attention outputs in a searched network and then perform retraining on the searched network using the training loss (2). Algorithm 1 in Section B of the appendix describes the search and retraining processes.

## 4 Experimental Results

We present the implementation details of our experiments in Section 4.1. We evaluate the performance of KCR-Transformers for image classification on the ImageNet-1k dataset in Section 4.2. In Section 4.3, we study the effectiveness of using KCR-Transformer as the feature extraction backbone for semantic segmentation and object detection. In Section 4.4, we study the effectiveness of the KCR-Transformer in reducing the KC of the networks. In Section 4.5, we examine the performance of KCR-Transformer as a vision encoder in vision-language models for the visual question answering tasks. In Section 4.6, we study the expected loss computed and the approximated KCR upper/lower bounds of the KCR-Transformers. Additional experiment results are presented in Section D of the appendix. In Section D.1, we evaluate the transferability of KCR-Transformer on downstream benchmarks. In Section D.2, we assess KCR-Transformer for self-supervised pretraining. In Section D.3, we describe the implementation details for instance segmentation. In Section D.4, we present the detailed object detection results. In Section D.5, we compare the training efficiency of KCR-Transformers with their corresponding baseline vision transformers and study the sensitivity to hyperparameters $\gamma_0$, $\eta$, and $m$. In Section D.6, we analyze the effect of varying $\lambda$ on the compression of the vision transformers by KCR. In Section D.7, we compare the inference time of KCR-Transformers with the competing baseline models. We present Grad-CAM visualizations results of KCR-Transformers in Section D.8.

## 4.1 EXPERIMENT SETTINGS

During the architecture search phase, we randomly sample a subset of 100 ImageNet classes (Russakovsky et al., 2015) for training. The network is optimized using the AdamW optimizer with a cosine learning rate schedule, where the initial learning rate of 0.001 is gradually annealed to 0.0001 over 200 epochs. In each epoch, 70% of the training samples are used for updating the model weights, while the remaining 30% are dedicated to optimizing the architecture parameters of the KCR-Transformer blocks. The temperature $\tau$ is initialized at 4.5 and decayed by a factor of 0.95 per epoch. Empirical results indicate that setting $t_{\text{warm}} = 90$ and $\eta = 1$ yields the best performance across all KCR model variants. During the retraining phase, the searched network is trained on the training set of ImageNet-1K using the AdamW optimizer with $\beta_1 = 0.9$ and $\beta_2 = 0.999$. We set $t_{\text{train}} = 300$ as the total number of training epochs, and $t_{\text{warm}} = 90$ for all the experiments. In addition, we search for the optimal values of feature rank $r$. Let $r = \lceil \gamma_0 \min(n, d) \rceil$, where $\gamma_0$ is the rank ratio. We select the value by performing 5-fold cross-validation on 20% of the training data. The value of $\gamma_0$ is selected from 0.1 to 0.5 with a step size of 0.05, and $\gamma_0 = 0.2$ is found to be the optimal. Throughout all the experiments, $m$ is set to 50000. $\lambda$ and $\eta$ are set to 0.2. As shown in Section D.5 of the appendix, the performance of the KCR-Transformer is stable with respect to different choices of the hyperparameters $\gamma_0, \eta, m$. The hyperparameter $\lambda$ controls the size of the searched architecture. Section D.6 shows the performance of KCR-Transformers compressed to variant sizes by setting $\lambda$ to different values. To improve the training efficiency, we compute $\tilde{\mathbf{U}}^{(r_0)}$, that is the approximation of $\mathbf{U}^{(r_0)}$, for every 30 epochs. Additional settings are presented in Section C of the appendix.

## 4.2 IMAGE CLASSIFICATION

We adopt ViT-S (Dosovitskiy et al., 2021), ViT-B (Dosovitskiy et al., 2021), Swin-T (Liu et al., 2021a), and Swin-B (Liu et al., 2021a) as backbone models. Each transformer block in these architectures is substituted with a KCR-Transformer block. For comparison, we evaluate the performance of KCR-Transformers against vision transformers compressed by the state-of-the-art kernel-based model compression method, NTK-SAP (Wang et al., 2023), and the state-of-the-art pruning method DeepCompress (Ahmed et al., 2025). Table 1 shows that KCR-Transformers consistently exhibit reduced computational costs and improved performance compared to the baseline vision transformers. For instance, KCR-Swin-B achieves a 1.1% gain in top-1 accuracy while reducing the computational cost by 2.8

| Model | # Params | FLOPs | Top-1 |
|---|---|---|---|
| T2T (Yuan et al., 2021) | 6.9 M | 1.8 G | 76.5 |
| PiT (Heo et al., 2021) | 10.6 M | 1.4 G | 78.1 |
| Mobile-Former (Chen et al., 2021e) | 9.4 M | 0.2 G | 76.7 |
| EViT (Liu et al., 2023b) | 12.4 M | 0.5 G | 77.1 |
| TinyViT (Wu et al., 2022) | 5.4 M | 1.3 G | 79.1 |
| EfficientFormer (Li et al., 2022b) | 12.3 M | 1.3 G | 79.2 |
| VTC-LFC (Wang et al., 2022) | 5.0 M | 1.3 G | 78.0 |
| SPViT (Kong et al., 2022) | 4.9 M | 1.2 G | 77.8 |
| EfficientViT-B1 (Cai et al., 2023) | 9.1 M | 0.52 G | 79.4 |
| MLP-Fusion-EfficientViT-B1 (Wei et al., 2023) | 7.9 M | 0.48 G | 79.1 |
| NTK-SAP-EfficientViT-B1 (Wang et al., 2023) | 8.0 M | 0.49 G | 79.4 |
| DeepCompress-EfficientViT-B1 (Ahmed et al., 2025) | 7.9 M | 0.46 G | 79.2 |
| **KCR-EfficientViT-B1** | 7.8 M | 0.44 G | **80.4** |
| ViT-S (Dosovitskiy et al., 2021) | 22.1 M | 4.3 G | 81.2 |
| MLP-Fusion-ViT-S (Wei et al., 2023) | 19.8 M | 4.0 G | 81.0 |
| NTK-SAP-ViT-S (Wang et al., 2023) | 20.3 M | 3.9 G | 80.9 |
| DeepCompress ViT-S (Ahmed et al., 2025) | 20.0 M | 3.9 G | 81.1 |
| **KCR-ViT-S (Ours)** | 19.8 M | 3.8 G | **82.2** |
| ViT-B (Dosovitskiy et al., 2021) | 86.5 M | 17.6 G | 83.7 |
| MLP-Fusion-ViT-B (Wei et al., 2023) | 70.2 M | 15.3 G | 83.5 |
| NTK-SAP-ViT-B (Wang et al., 2023) | 71.8 M | 15.6 G | 83.5 |
| DeepCompress ViT-B (Ahmed et al., 2025) | 70.5 M | 15.1 G | 83.6 |
| **KCR-ViT-B (Ours)** | 69.5 M | 14.5 G | **84.6** |
| Swin-T (Liu et al., 2021a) | 29.0 M | 4.5 G | 81.3 |
| MLP-Fusion-Swin-T (Wei et al., 2023) | 24.8 M | 4.1 G | 81.0 |
| NTK-SAP-Swin-T (Wang et al., 2023) | 25.5 M | 4.2 G | 81.2 |
| DeepCompress Swin-T (Ahmed et al., 2025) | 24.8 M | 4.1 G | 81.1 |
| **KCR-Swin-T (Ours)** | 24.6 M | 3.9 G | **82.4** |
| Swin-B (Liu et al., 2021a) | 88.0 M | 15.4 G | 83.5 |
| MLP-Fusion-Swin-B (Wei et al., 2023) | 70.8 M | 13.3 G | 83.2 |
| NTK-SAP-Swin-B (Wang et al., 2023) | 72.6 M | 13.2 G | 83.2 |
| DeepCompress Swin-B (Ahmed et al., 2025) | 71.5 M | 13.0 G | 83.1 |
| **KCR-Swin-B (Ours)** | 70.2 M | 12.6 G | **84.7** |

Table 1: Performance comparison on ImageNet-1k.

GFLOPs, highlighting the effectiveness of KCR in enhancing both efficiency and predictive performance. Notably, KCR-EfficientViT-B1 achieves the lowest computational cost among all evaluated models, requiring only 0.44 GFLOPs while even outperforming its uncompressed counterpart by 1.0% and all baselines in the first part of Table 1 (T2T-SPViT) in top-1 accuracy, demonstrating the effectiveness of KCR-Transformers for resource-constrained applications. Moreover, Table 9 in the appendix shows that the performance and efficiency improvements by the KCR-Transformer are achieved with only marginal increases ($< 7.8\%$) in the training time. Table 14 shows that KCR-Transformers exhibit faster inference speed than models compressed by competing baselines.

### 4.3 OBJECT DETECTION AND SEMANTIC SEGMENTATION

We incorporate the ImageNet pre-trained KCR-Swin-B into the Cascade Mask R-CNN framework (Cai & Vasconcelos, 2021) for object detection. All models are evaluated on the MS-COCO dataset (Lin et al., 2014). The implementation details for the experiments are presented in Section D.4 of the appendix. It is observed in Table 2 that compressing the Swin backbones with KCR consistently improves both box-level and mask-level detection performance within the Cascade Mask R-CNN framework. For example, KCR-Swin-B achieves a box mAP of $52.5\%$ and a mask mAP of $45.6\%$, with improvements of $0.6\%$ and $0.6\%$ over the standard Swin-B baseline. Results with AP at IoU thresholds of $50$ and $75$ are deferred to Table 8 in Section D.4 of the appendix.

In addition, we evaluate the performance of KCR for segmentation on the ADE20K (Zhou et al., 2019) using UperNet (Xiao et al., 2018) with our KCR-Swin-B as the feature extraction backbone. We include Swin-B (Liu et al., 2021a) and SETR (Zheng et al., 2021) as baselines for comparisons. We follow the training and evaluation protocol in (Liu et al., 2021a), where both our model and the baselines are trained on the training split and evaluated on the validation split of the dataset. More implementation details for the experiments are presented in Section D.3. As shown in Table 3, UperNet equipped with our KCR-Swin-B backbone achieves the highest validation mIoU of $52.4\%$, surpassing UperNet with Swin-B by $0.8\%$ and SETR with ViT-L by $2.1\%$. These performance gains further highlight the effectiveness of the low-rank regularization and kernel complexity reduction introduced by KCR in enhancing the representational capacity of the vision backbone.

| Framework | Feature Backbone | mAP$^{box}$ | mAP$^m$ |
|---|---|---|---|
| Mask R-CNN | Swin-B | 51.9 | 45.0 |
| Mask R-CNN | NTK-SAP-Swin-B | 51.5 | 44.6 |
| Mask R-CNN | KCR-Swin-B (Ours) | **52.5** | **45.6** |

Table 2: Object Detection Results on COCO.

| Framework | Feature Backbone | Val mIoU |
|---|---|---|
| SETR | ViT-L | 50.3 |
| UperNet | Swin-B | 51.6 |
| UperNet | NTK-SAP-Swin-B | 51.3 |
| UperNet | KCR-Swin-B (Ours) | **52.4** |

Table 3: Segmentation Results on ADE20K.

### 4.4 ABLATION STUDY ON THE EFFECTS OF KCR-TRANSFORMER IN REDUCING THE KC

We conduct an ablation study to examine the effectiveness of KCR-Transformer in reducing the KC across four widely used vision transformer architectures, including ViT-S, ViT-B, Swin-T, and Swin-B. For each model, we compare the vanilla version against its KCR-Transformer counterpart. As shown in Table 4, the KCR-enhanced models consistently exhibit a substantial reduction in KC while simultaneously achieving lower parameter counts, reduced FLOPs, and higher top-1 classification accuracy on ImageNet-1K. For instance, KCR-ViT-S reduces the KC from $4.12$ to $0.65$ and improves the top-1 accuracy from $81.2\%$ to $82.2\%$ with fewer FLOPs and parameters. These improvements demonstrate that our approximate TNN regularization effectively reduces the

| Model | # Params | FLOPs | Top-1 | KC |
|---|---|---|---|---|
| EfficientViT-B1 (Cai et al., 2023) | 9.1 M | 0.52 G | 79.4 | 4.08 |
| MLP-Fusion-EfficientViT-B1 (Wei et al., 2023) | 7.9 M | 0.48 G | 79.1 | 3.28 |
| NTK-SAP-EfficientViT-B1 (Ahmed et al., 2025) | 8.0 M | 0.49 G | 79.4 | 3.17 |
| DeepCompress-EfficientViT-B1 (Cai et al., 2023) | 7.9 M | 0.46 G | 79.2 | 3.68 |
| **KCR-EfficientViT-B1** (Cai et al., 2023) | 7.8 M | 0.44 G | **80.4** | **0.62** |
| ViT-S (Dosovitskiy et al., 2021) | 22.1 M | 4.3 G | 81.2 | 4.12 |
| MLP-Fusion-ViT-S (Wei et al., 2023) | 19.8 M | 4.0 G | 81.0 | 3.88 |
| NTK-SAP-ViT-S (Wang et al., 2023) | 20.3 M | 3.9 G | 80.9 | 3.25 |
| DeepCompress ViT-S (Ahmed et al., 2025) | 20.0 M | 3.9 G | 81.1 | 3.97 |
| **KCR-ViT-S (Ours)** | 19.8 M | 3.8 G | **82.2** | **0.65** |
| ViT-B (Dosovitskiy et al., 2021) | 86.5 M | 17.6 G | 83.7 | 4.35 |
| MLP-Fusion-ViT-B (Wei et al., 2023) | 70.2 M | 15.3 G | 83.5 | 3.62 |
| NTK-SAP-ViT-B (Wang et al., 2023) | 71.8 M | 15.6 G | 83.5 | 3.58 |
| DeepCompress ViT-B (Ahmed et al., 2025) | 70.5 M | 15.1 G | 83.6 | 3.85 |
| **KCR-ViT-B (Ours)** | 69.5 M | 14.5 G | **84.6** | **0.52** |
| Swin-T (Liu et al., 2021a) | 29.0 M | 4.5 G | 81.3 | 3.42 |
| MLP-Fusion-Swin-T (Wei et al., 2023) | 24.8 M | 4.1 G | 81.0 | 3.10 |
| NTK-SAP-Swin-T (Wang et al., 2023) | 25.5 M | 4.2 G | 81.2 | 1.86 |
| DeepCompress Swin-T (Ahmed et al., 2025) | 24.8 M | 4.1 G | 81.1 | 2.42 |
| **KCR-Swin-T (Ours)** | 24.6 M | 3.9 G | **82.4** | **0.44** |
| Swin-B (Liu et al., 2021a) | 88.0 M | 15.4 G | 83.5 | 3.21 |
| MLP-Fusion-Swin-B (Wei et al., 2023) | 70.8 M | 13.3 G | 83.4 | 2.60 |
| NTK-SAP-Swin-B (Wang et al., 2023) | 72.6 M | 13.2 G | 83.2 | 2.42 |
| DeepCompress Swin-B (Ahmed et al., 2025) | 71.5 M | 13.0 G | 83.4 | 2.68 |
| **KCR-Swin-B (Ours)** | 70.2 M | 12.6 G | **84.7** | **0.41** |

Table 4: Ablation Study on the Effects of KCR-Transformer in Reducing the KC.

KC of the vision backbones, leading to improved generalization capability. In contrast, the existing kernel-based compression method NTK-SAP (Wang et al., 2023) is limited by the conventional NTK limit, so it does not improve the top-1 accuracy of the pruned models, while its KC is slightly smaller than that of the original model as a result of pruning. MLP-Fusion-ViT-B (Wei et al., 2023) and DeepCompress ViT-B (Ahmed et al., 2025) also renders compressed models with worse top-1 accuracy and slightly reduced KC compared to the original model, due to the compression effect.

### 4.5 VISUAL QUESTION ANSWERING (VQA) WITH KCR-TRANSFORMER AS THE VISION ENCODER IN VISION-LANGUAGE MODELS (VLMS)

We further evaluate the KCR-Transformer in the visual question answering (VQA) task by employing it as the vision encoder within the vision-language model (VLM) framework, FLAVA (Singh et al., 2022). The evaluation is performed on widely used VQA datasets VQAv2 (Goyal et al., 2017) and SNLI-VE (Xie et al., 2019). We replace the vision encoder, ViT-B, in the VLM framework, FLAVA, with our KCR-Transformer during the vision encoder pre-training process. For VQAv2, we follow the settings in FLAVA (Singh et al., 2022) and fine-tune the model using a cross-entropy loss over the top frequent answer candidates. For SNLI-VE, we also follow the settings in FLAVA (Singh et al., 2022) and fine-tune the model using a standard three-way classification loss to distinguish entailment, contradiction, and neutrality. The test-dev VQA score for VQAv2 and the test accuracy for SNLI-VE are reported following (Singh et al., 2022). It is observed in Table 5 that FLAVA with KCR-ViT-B consistently outperforms the original FLAVA with ViT-B and the baseline model with ViT-B compressed by NTK-SAP across both benchmarks while maintaining a lower computational cost. For example, FLAVA with KCR-ViT-B achieves a VQAv2 score of 73.26%, outperforming FLAVA with ViT-B by 0.77% and FLAVA with NTK-SAP-ViT-B by 1.18%.

| Model | # Params | FLOPs | VQAv2 | SNLI-VE |
|---|---|---|---|---|
| FLAVA (ViT-B) (Dosovitskiy et al., 2021) | 86.5 M | 17.6 G | 72.49 | 78.89 |
| FLAVA (NTK-SAP-ViT-B) (Wang et al., 2023) | 71.8 M | 15.6 G | 72.08 | 78.16 |
| FLAVA (KCR-ViT-B) | 69.5 M | 14.5 G | **73.26** | **79.35** |

Table 5: Performance for VQA on VQAv2 (Goyal et al., 2017) and SNLI-VE (Xie et al., 2019).

### 4.6 STUDY ON THE TIGHTNESS OF THE KCR UPPER AND LOWER BOUNDS

Figure 1 illustrates the expected loss computed over the training and validation sets of the ImageNet-1K and the approximated KCR upper/lower bounds computed at different training epochs for ViT-S, ViT-B, Swin-T, and Swin-B trained on the ImageNet-1K dataset. We note that the approximated KCR upper/lower bounds are the KCR upper/lower bounds with the KC replaced by the approximate KC, A-KC, which is defined as $\text{A-KC}(\mathbf{K}) \coloneqq \min_{h \in [0, r_0]} \left( \frac{h}{n} + \sqrt{\frac{1}{n} \overline{\|\mathbf{K}\|}_h} \right)$. Since the approximate TNN $\overline{\|\mathbf{K}\|}_h$ is expected to be close to the TNN $\|\mathbf{K}\|_h$ for each $h \in [0\colon r_0]$, the approximate KC A-KC($\mathbf{K}$) is also expected to be close to the KC, KC($\mathbf{K}$). It can be observed that the approximated KCR upper/lower bounds are tightly correlated to the expected loss, revealing the tightness of the upper/lower bounds for the generalization error of the DNNs with KCR-Transformer blocks.

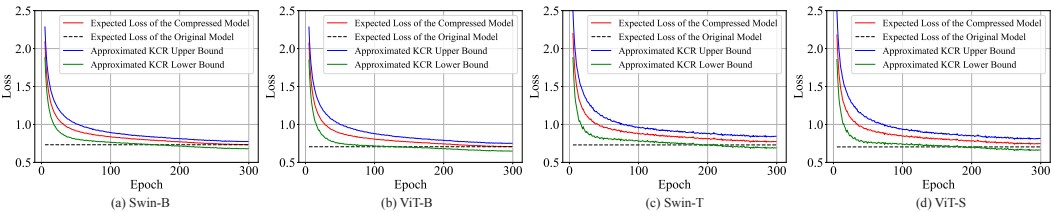

Figure 1: Illustration of the expected loss and the approximated KCR upper/lower bounds over different training epochs for ViT-S, ViT-B, Swin-T, and Swin-B compressed by KCR.

## 5 CONCLUSION

In this paper, we propose KCR-Transformer, a novel and generalization-aware transformer block equipped with differentiable channel selection for the MLP layers in vision transformers. Guided by a novel and sharp theoretical generalization bound derived from the kernel complexity (KC) of the network, KCR-Transformer enables channel pruning in a theoretically grounded and principled manner. Our method is compatible with a wide range of vision transformer architectures and can be seamlessly integrated to replace standard transformer blocks. Extensive experiments across image classification, object detection, and semantic segmentation demonstrate that KCR-Transformer consistently achieves superior performance with fewer FLOPs and parameters, validating its effectiveness for building efficient vision transformers.

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

## A  PROOF OF THEOREM 3.1

**Proof of Theorem 3.1.** We consider the dynamic kernel $K(\mathbf{x}, \mathbf{x}') = F(\mathbf{x})^\top F(\mathbf{x}')$ for all $\mathbf{x}, \mathbf{x}' \in \mathcal{X}$ defined in Section 3.2. Then it follows from (Bartlett et al., 2005, Theorem 3.3) that for every $K > 1$,

$$L_{\mathcal{D}}(\mathrm{NN}_{\mathcal{W}}) \leq \frac{K}{K-1} \mathbb{E}_{P_n} \left[ \|\mathrm{NN}_{\mathcal{W}}(\mathbf{x}) - \mathbf{y}\|_2^2 \right] + \Theta(r^*) + \Theta\left(\frac{x}{n}\right),$$

$$L_{\mathcal{D}}(\mathrm{NN}_{\mathcal{W}}) \geq \frac{K}{K+1} \mathbb{E}_{P_n} \left[ \|\mathrm{NN}_{\mathcal{W}}(\mathbf{x}) - \mathbf{y}\|_2^2 \right] - \Theta(r^*) - \Theta\left(\frac{x}{n}\right), \tag{3}$$

and each inequality in (3) holds with probability at least $1 - \exp(-x)$. $K$ is a positive definite kernel and the network $\mathrm{NN}_{\mathcal{W}} \in \mathcal{H}_K$ where $\mathcal{H}_K$ denotes the Reproducing Kernel Hilbert Space

(RKHS) associated with $K$, and we recall that $\text{KC}(\mathbf{K}) = \min\limits_{h \in [0, r_0]} \left( \frac{h}{n} + \sqrt{\frac{1}{n} \sum\limits_{i=h+1}^{r_0} \widehat{\lambda}_i} \right)$ is the kernel complexity. It then follows from (Bartlett et al., 2005, Corollary 6.7) and (3) that

$$L_{\mathcal{D}}(\text{NN}_{\mathcal{W}}) \le \frac{K}{K-1} \mathbb{E}_{P_n} \left[ \|\text{NN}_{\mathcal{W}}(\mathbf{x}) - \mathbf{y}\|_2^2 \right] + \Theta(\text{KC}(\mathbf{K})) + \Theta \left( \frac{x}{n} \right),$$

$$L_{\mathcal{D}}(\text{NN}_{\mathcal{W}}) \ge \frac{K}{K+1} \mathbb{E}_{P_n} \left[ \|\text{NN}_{\mathcal{W}}(\mathbf{x}) - \mathbf{y}\|_2^2 \right] - \Theta(\text{KC}(\mathbf{K})) - \Theta \left( \frac{x}{n} \right). \tag{4}$$

Combining (3) and (4) proves (1).

$\square$

# B  ALGORITHMS

Algorithm 1 describes the search and the retraining process of the KCR-Transformer.

---
**Algorithm 1** Training Algorithm with the Approximate Truncated Nuclear Norm by SGD

---
**Require:** Training dataset $\mathcal{D}$, number of search epochs $t_{\text{search}}$, number of training epochs $t_{\text{train}}$, warm-up epochs $t_{\text{warm}}$, batch size $J$, learning rates $\eta_{\mathcal{W}}$ and $\eta_\alpha$

**Ensure:** Trained weights $\mathcal{W}$ of the network

1: Split $\mathcal{D}$ into $\mathcal{D}_{\mathcal{W}}$ and $\mathcal{D}_\alpha$, where 70% of samples ($\mathcal{D}_{\mathcal{W}}$) are used to update model weights $\mathcal{W}$, and 30% of samples ($\mathcal{D}_\alpha$) are used to optimize architecture parameters $\alpha$.
2: Initialize the network weights $\mathcal{W} = \mathcal{W}(0)$ by random initialization.
3: **for** $t \leftarrow 1$ to $t_{\text{search}}$ **do**
4:     **for** $j \leftarrow 1$ to $J$ **do**
5:         Sample mini-batches $\mathcal{B}_j^{\mathcal{W}} \subset \mathcal{D}_{\mathcal{W}}, \mathcal{B}_j^\alpha \subset \mathcal{D}_\alpha$.
6:         Perform gradient descent on $\mathcal{B}_j^\alpha$ to update $\alpha$ by $\alpha \leftarrow \alpha - \eta_\alpha \nabla_\alpha \mathcal{L}_{\text{search},j}^{(t)}(\mathcal{W}, \alpha)$
7:         Perform gradient descent on $\mathcal{B}_j^{\mathcal{W}}$ to update $\mathcal{W}$ by $\mathcal{W} \leftarrow \mathcal{W} - \eta_{\mathcal{W}} \nabla_{\mathcal{W}} \mathcal{L}_{\text{search},j}^{(t)}(\mathcal{W}, \alpha)$
8:     **end for**
9: **end for**
10: **for** $t \leftarrow 1$ to $t_{\text{train}}$ **do**
11:     **if** $t \bmod 30 = 0$ **then**
12:         Update $\tilde{\mathbf{U}}^{(r_0)}$, the approximation of $\mathbf{U}^{(r_0)}$.
13:     **end if**
14:     **for** $j \leftarrow 1$ to $J$ **do**
15:         Sample mini-batch $\mathcal{B}_j \subset \mathcal{D}_{\mathcal{W}}$.
16:         **if** $t > t_{\text{warm}}$ **then**
17:             Perform gradient descent on $\mathcal{B}_j^{\mathcal{W}}$ to update $\mathcal{W}$ by $\mathcal{W} \leftarrow \mathcal{W} - \eta_{\mathcal{W}} \nabla_{\mathcal{W}} \mathcal{L}_{\text{train},j}^{(t)}(\mathcal{W})$
18:         **else**
19:             Perform gradient descent on $\mathcal{B}_j^{\mathcal{W}}$ to update $\mathcal{W}$ by $\mathcal{W} \leftarrow \mathcal{W} - \eta_{\mathcal{W}} \nabla_{\mathcal{W}} \text{CE}_j^{(t)}(\mathcal{W})$
20:         **end if**
21:     **end for**
22: **end for**
23: **return** $\mathcal{W}$

---

# C  IMPLEMENTATION DETAILS

All experiments are conducted on 40GB NVIDIA A100 GPUs with an effective batch size of 512. Following standard practice (Cai et al., 2023), we apply widely adopted data augmentation techniques during training, including random scaling, random horizontal flipping, and random cropping. The weight decay is set to 0.01. The learning rate is linearly increased from 0.0002 to 0.002 over the first five epochs, then gradually annealed back to 0.0002 using a cosine decay schedule over the remaining epochs. Inference is performed using the exponential moving average (EMA) of model weights.

# D ADDITIONAL EXPERIMENT RESULTS

## D.1 TRANSFER LEARNING CAPABILITY OF THE KCR-TRANSFORMER

We evaluate the transfer learning capability of KCR-ViT-B in comparison to the baseline ViT-B using three widely adopted benchmarks, including Oxford Flowers-102 (Nilsback & Zisserman, 2008), Oxford-IIIT Pet (Parkhi et al., 2012), and Stanford Cars (Krause et al., 2013). Following the established transfer learning protocol in (Kolesnikov et al., 2020), both models are first pre-trained on ImageNet and subsequently fine-tuned on the respective training sets of the target datasets for 50 epochs using the Adam optimizer, with a fixed learning rate of $1 \times 10^{-5}$. The experimental results are presented in Table 6. It is observed that KCR-ViT-B consistently outperforms ViT-B and NTK-SAP-ViT-B across all three datasets, while requiring fewer FLOPs and model parameters. For example, KCR-ViT-B achieves a top-1 accuracy of $93.8\%$, outperforming ViT-B and NTK-SAP-ViT-B by $1.1\%$ and $1.3\%$, respectively, on the Cars dataset. The improvements demonstrate the superior transferability of the features learned by the KCR compressed model across diverse visual domains.

| Model | # Params | FLOPs | ImageNet | Flowers | Pet | Cars |
|---|---|---|---|---|---|---|
| ViT-B (Dosovitskiy et al., 2021) | 86.5 M | 17.6 G | 83.7 | 97.8 | 96.0 | 92.7 |
| NTK-SAP-ViT-B (Wang et al., 2023) | 71.8 M | 15.6 G | 83.5 | 97.4 | 95.6 | 92.5 |
| KCR-ViT-B | 69.5 M | 14.5 G | **84.6** | **98.2** | **96.6** | **93.8** |

Table 6: Top-1 classification accuracy comparison for transfer learning on the Oxford Flowers-102 (Nilsback & Zisserman, 2008), Oxford-IIIT Pet (Parkhi et al., 2012), and Stanford Cars (Krause et al., 2013) datasets.

## D.2 SELF-SUPERVISED LEARNING WITH KCR-TRANSFORMERS

We further evaluate the effectiveness of KCR-Transformers in the self-supervised learning (SSL) setting using both MoCoV3 (Chen et al., 2021d) and MOCA (Gidaris et al., 2024). In our experiments, the ViT-B model pre-trained with each SSL method serves as the baseline. We pre-train ViT-B, NTK-SAP-ViT-B, and our KCR-ViT-B strictly following the training settings described in the respective papers (Chen et al., 2021d; Gidaris et al., 2024) on the ImageNet1K without using training labels. The pre-trained models are then subsequently fine-tuned with class labels for downstream classification on ImageNet1K following the fine-tuning protocols in (Chen et al., 2021d; Gidaris et al., 2024). As shown in Table 7, KCR-ViT-B consistently achieves superior performance over ViT-B and NTK-SAP-ViT-B across both SSL pipelines, while requiring fewer FLOPs and parameters. For example, under the MOCA (Gidaris et al., 2024) pretraining pipeline, KCR-ViT-B achieves a top-1 accuracy of $84.4\%$, outperforming ViT-B and NTK-SAP-ViT-B by $1.0\%$ and $1.2\%$, respectively, with reduced computational cost and parameter size. The improvements highlight the effectiveness of the KCR compression method in enhancing the efficiency and accuracy of the models pre-trained by SSL methods.

| Network | SSL Method | # Params | FLOPs | Top-1 |
|---|---|---|---|---|
| ViT-B (Dosovitskiy et al., 2021) | | 86.5 M | 17.6 G | 83.2 |
| NTK-SAP-ViT-B (Wang et al., 2023) | MoCoV3 (Chen et al., 2021d) | 71.8 M | 15.6 G | 82.9 |
| KCR-ViT-B (Ours) | | 69.5 M | 14.5 G | **84.1** |
| ViT-B (Dosovitskiy et al., 2021) | | 86.5 M | 17.6 G | 83.4 |
| NTK-SAP-ViT-B (Wang et al., 2023) | MOCA (Gidaris et al., 2024) | 71.8 M | 15.6 G | 83.2 |
| KCR-ViT-B (Ours) | | 69.5 M | 14.5 G | **84.4** |

Table 7: Top-1 classification accuracy comparison for models pre-trained with the self-supervised learning (SSL) methods MoCoV3 (Chen et al., 2021d) and MOCA (Gidaris et al., 2024) on ImageNet1K.

## D.3 IMPLEMENTATION DETAILS FOR INSTANCE SEGMENTATION

ADE20K has 25000 images in total, with 20000 for training, 2000 for validation, and another 3000 for testing. We adopt UperNet (Xiao et al., 2018) as the segmentation framework with our KCR-

Swin-B as the feature extraction backbone. We follow the training and evaluation protocol in (Liu et al., 2021a), where both our model and the baselines are trained on the training split and evaluated on the validation split of the dataset. All models are optimized using AdamW for a total of $160000$ iterations with a batch size of $16$, an initial learning rate of $6 \times 10^{-5}$, and a weight decay of $0.01$. The learning rate follows a linear decay schedule after a warm-up phase of $1500$ iterations. To enhance generalization, we employ data augmentation techniques including random horizontal flipping, random rescaling with a ratio range of $[0.5, 2.0]$, and random photometric distortions. Stochastic depth regularization is applied with a drop rate of $0.2$. For inference, we use multi-scale testing with scale factors varying from $0.5$ to $1.75$.

### D.4 IMPLEMENTATION DETAILS AND ADDITIONAL RESULTS FOR OBJECT DETECTION

We incorporate the ImageNet pre-trained KCR-Swin-T and KCR-Swin-B into the Cascade Mask R-CNN framework (Cai & Vasconcelos, 2021) for object detection. All models are evaluated on the MS-COCO dataset (Lin et al., 2014), which consists of $117000$ training images and $5000$ validation images. We follow the training configuration of (Liu et al., 2021a), where each input image is resized such that the shorter side falls within $[480, 800]$ pixels while the longer side does not exceed $1333$ pixels. Training is performed using the AdamW optimizer with an initial learning rate of $0.0001$, a weight decay of $0.05$, and a batch size of $16$, for a total of $36$ epochs following the $3\times$ schedule. In line with (Cai & Vasconcelos, 2021), we report standard COCO metrics, including the box-level mean Average Precision (mAP$^{box}$) and mask-level mean Average Precision (mAP$^m$), as well as AP at IoU thresholds of $50$ and $75$.

| Detection Framework | Feature Backbone | mAP$^{box}$ | AP$_{50}^b$ | AP$_{75}^b$ | mAP$^m$ | AP$_{50}^m$ | AP$_{75}^m$ |
|---|---|---|---|---|---|---|---|
| Mask R-CNN | Swin-T | 50.5 | 69.3 | 54.9 | 43.7 | 66.6 | 47.1 |
| Mask R-CNN | Swin-B | 51.9 | 70.9 | 56.5 | 45.0 | 68.4 | 48.7 |
| Mask R-CNN | KCR-Swin-T (Ours) | 50.9 | 69.7 | 55.3 | 44.0 | 67.1 | 47.6 |
| Mask R-CNN | KCR-Swin-B (Ours) | 52.5 | 71.4 | 56.8 | 45.6 | 68.9 | 49.1 |

Table 8: Detailed Object Detection Results on COCO.

It is observed in Table 8 that compressing the Swin backbones with KCR consistently improves both box-level and mask-level detection performance within the Cascade Mask R-CNN framework. For example, KCR-Swin-T achieves a box mAP of $50.9\%$ and a mask mAP of $44.0\%$, with improvements of $0.4\%$ and $0.3\%$ over the standard Swin-T baseline. Similarly, KCR-Swin-B achieves the highest box mAP of $52.5\%$ and mask mAP of $45.6\%$, surpassing the Swin-B baseline by $0.6\%$ and $0.6\%$, respectively. These results demonstrate that KCR compression effectively enhances the feature expressiveness of vision backbones for object detection, with even less computational costs. The consistent improvements across multiple IoU thresholds further validate the robustness and generalization capability of vision backbones compressed by KCR.

### D.5 TRAINING EFFICIENCY AND SENSITIVITY OF HYPERPARAMETERS $\gamma_0, \eta, m$

In this section, we first compare the training time of the KCR-Transformers with the corresponding baseline vision transformers without the KCR regularization. The comparison is performed on one NVIDIA A100 40GB GPU with full-precision, and the average training time per epoch is reported. It is observed from Table 9 that the KCR regularization only marginally increases the training time ($< 7.8\%$) compared to the corresponding baseline vision transformers, while greatly increasing the top-1 classification accuracy and reducing the model size of FLOPs. For example, KCR-Swin-B achieves a $1.2\%$ top-1 accuracy improvement while reducing the number of parameters from $88.0$M to $70.2$M and FLOPs from $15.4$G to $12.6$G, with a negligible training overhead of $1.6$ minutes per epoch, which is only $4.18\%$ of the original training time of Swin-B. In addition, we also report the overall search time and re-training time in comparison with the training time of the baseline model. It is observed that the search phase only brings marginal training overhead, costing less than $7.2\%$ of the baseline model's training time.

We further study the sensitivity of KCR-Transformer to the key hyperparameters $\gamma_0$, $\eta$, and $m$ by conducting experiments on the Swin-B backbone with different values of $\gamma_0$, $\eta$, and $m$. When

| Model | # Params | FLOPs | Top-1 | Training Efficiency (Minutes/Epoch) | Search Time (Hours) | Re-Training Time (Hours) |
|---|---|---|---|---|---|---|
| ViT-S (Dosovitskiy et al., 2021) | 22.1 M | 4.3 G | 81.2 | 16.8 | - | 84.0 |
| **KCR-ViT-S (Ours)** | 19.8 M | 3.8 G | **82.2** | 18.1 | 6.03 | 88.5 |
| ViT-B (Dosovitskiy et al., 2021) | 86.5 M | 17.6 G | 83.7 | 33.5 | - | 167.5 |
| **KCR-ViT-B (Ours)** | 69.5 M | 14.5 G | **84.6** | 35.2 | 11.7 | 173.4 |
| Swin-T (Liu et al., 2021a) | 29.0 M | 4.5 G | 81.3 | 20.8 | - | 104.0 |
| **KCR-Swin-T (Ours)** | 24.6 M | 3.9 G | **82.4** | 22.1 | 7.3 | 108.5 |
| Swin-B (Liu et al., 2021a) | 88.0 M | 15.4 G | 83.5 | 38.3 | - | 191.5 |
| **KCR-Swin-B (Ours)** | 70.2 M | 12.6 G | **84.7** | 39.9 | 13.3 | 197.1 |

Table 9: Training time comparisons with baseline methods on the training set of ImageNet-1K.

performing the study on one of the hyperparameters, the remaining hyperparameters are set to their corresponding optimal values. The results summarized in Tables 10, 11, and 12. Across all settings, the top-1 accuracy remains stable within an error range of $0.3\%$, indicating that KCR-Transformer is robust to the choice of these hyperparameters.

| $\gamma_0$ | 0.1 | 0.2 | 0.3 | 0.4 | 0.5 |
|---|---|---|---|---|---|
| Top-1 (%) | 84.6 | 84.7 | 84.5 | 84.7 | 84.6 |

Table 10: Sensitivity analysis of KCR-Swin-B to the choices of $\gamma_0$ on the ImageNet-1K dataset.

| $\eta$ | 0.1 | 0.2 | 0.3 | 0.4 | 0.5 |
|---|---|---|---|---|---|
| Top-1 (%) | 84.6 | 84.7 | 84.7 | 84.6 | 84.5 |

Table 11: Sensitivity analysis of KCR-Swin-B to the choices of $\eta$ on the ImageNet-1K dataset.

| $m$ | 25000 | 50000 | 75000 | 100000 | 200000 |
|---|---|---|---|---|---|
| Top-1 (%) | 84.4 | 84.7 | 84.6 | 84.7 | 84.7 |

Table 12: Sensitivity analysis of KCR-Swin-B to the choices of $m$ on the ImageNet-1K dataset.

## D.6 KCR-TRANSFORMERS WITH DIFFERENT COMPRESSION RATIOS

In this section, we investigate the performance of KCR-Transformer under different compression ratios by varying the hyperparameter $\lambda$ in the search loss. Larger values of $\lambda$ impose stronger penalization on the computational cost, thereby yielding more aggressively compressed architectures. We evaluate four representative settings, $\lambda \in \{0.2, 0.4, 0.6, 0.8\}$, using the Swin-B backbone on the ImageNet-1K dataset. In addition, we compare KCR-Swin-B with Swin-B compressed by NTK-SAP (Wang et al., 2023) of similar sizes. As shown in Table 13, KCR-Transformer consistently outperforms NTK-SAP across all compression levels. Notably, for each comparable model size, KCR-Swin-B achieves significantly higher top-1 accuracy while maintaining lower parameter count and FLOPs. For example, at $\lambda = 0.2$, KCR-Swin-B achieves $84.7\%$ top-1 accuracy with only 12.6 G FLOPs, outperforming NTK-SAP-Swin-B ($82.8\%$, 10.2 G FLOPs) by $1.9\%$ in top-1 accuracy. This trend persists even under more aggressive compression settings, such as $\lambda = 0.8$, where KCR-Swin-B achieves a top-1 accuracy of $82.9\%$, despite having fewer parameters and FLOPs than its NTK-SAP counterpart.

## D.7 INFERENCE TIME COMPARISON

In this section, we further compare the inference latency of the KCR-Transformers against their corresponding baseline vision transformers without the KCR regularization. All measurements are conducted on two NVIDIA A100 40GB GPUs using FP16 precision with a batch size of 128. As shown in Table 14, KCR-Transformer variants consistently achieve lower inference latency compared to their counterparts, while simultaneously improving top-1 classification accuracy and reducing both the parameter count and computational cost (FLOPs). For instance, KCR-Swin-B achieves a $1.2\%$ gain in top-1 accuracy over Swin-B, while reducing the number of parameters from 88.0M

| Model | $\lambda$ | # Params (M) | FLOPs (G) | Top-1 (%) |
|---|---|---|---|---|
| Swin-B | - | 88.0 | 15.4 | 83.5 |
| NTK-SAP-Swin-B (Wang et al., 2023) | - | 72.6 | 13.2 | 83.2 |
| KCR-Swin-B | 0.2 | 70.2 | 12.6 | **84.7** |
| NTK-SAP-Swin-B (Wang et al., 2023) | - | 57.9 | 10.2 | 82.8 |
| KCR-Swin-B | 0.4 | 55.4 | 9.7 | **84.2** |
| NTK-SAP-Swin-B (Wang et al., 2023) | - | 49.3 | 8.4 | 82.2 |
| KCR-Swin-B | 0.6 | 46.7 | 7.8 | **83.4** |
| NTK-SAP-Swin-B (Wang et al., 2023) | - | 40.2 | 6.3 | 81.5 |
| KCR-Swin-B | 0.8 | 39.5 | 6.0 | **82.9** |

Table 13: Performance of KCR-Swin-B under different compression ratios. Increasing $\lambda$ enforces stronger compression, leading to smaller parameter sizes and FLOPs.

Table 14: Comparisons with baseline methods on ImageNet-1K validation set (inference time measured on $2\times$ NVIDIA A100 40GB GPU, in the precision of FP16, with a batch size = 128).

| Model | # Params | FLOPs | Top-1 | Inference Time (ms/img) |
|---|---|---|---|---|
| EfficientViT-B1 (Cai et al., 2023) | 9.1 M | 0.52 G | 79.4 | 0.312 |
| MLP-Fusion-EfficientViT-B1 (Wei et al., 2023) | 7.9 M | 0.48 G | 79.1 | 0.298 |
| NTK-SAP-EfficientViT-B1 (Ahmed et al., 2025) | 8.0 M | 0.49 G | 79.4 | 0.304 |
| DeepCompress-EfficientViT-B1 (Cai et al., 2023) | 7.9 M | 0.46 G | 79.2 | 0.285 |
| **KCR-EfficientViT-B1** (Cai et al., 2023) | 7.8 M | 0.44 G | **80.4** | **0.271** |
| ViT-S (Dosovitskiy et al., 2021) | 22.1 M | 4.3 G | 81.2 | 0.642 |
| MLP-Fusion-ViT-S (Wei et al., 2023) | 19.8 M | 4.0 G | 81.0 | 0.596 |
| NTK-SAP-ViT-S (Wang et al., 2023) | 20.3 M | 3.9 G | 80.9 | 0.583 |
| DeepCompress ViT-S (Ahmed et al., 2025) | 20.0 M | 3.9 G | 81.1 | 0.574 |
| **KCR-ViT-S (Ours)** | 19.8 M | 3.8 G | **82.2** | **0.561** |
| ViT-B (Dosovitskiy et al., 2021) | 86.5 M | 17.6 G | 83.7 | 1.594 |
| MLP-Fusion-ViT-B (Wei et al., 2023) | 70.2 M | 15.3 G | 83.5 | 1.442 |
| NTK-SAP-ViT-B (Wang et al., 2023) | 71.8 M | 15.6 G | 83.5 | 1.467 |
| DeepCompress ViT-B (Ahmed et al., 2025) | 70.5 M | 15.1 G | 83.6 | 1.426 |
| **KCR-ViT-B (Ours)** | 69.5 M | 14.5 G | **84.6** | **1.378** |
| Swin-T (Liu et al., 2021a) | 29.0 M | 4.5 G | 81.3 | 0.673 |
| MLP-Fusion-Swin-T (Wei et al., 2023) | 24.8 M | 4.1 G | 81.0 | 0.631 |
| NTK-SAP-Swin-T (Wang et al., 2023) | 25.5 M | 4.2 G | 81.2 | 0.644 |
| DeepCompress Swin-T (Ahmed et al., 2025) | 24.8 M | 4.1 G | 81.1 | 0.622 |
| **KCR-Swin-T (Ours)** | 24.6 M | 3.9 G | **82.4** | **0.598** |
| Swin-B (Liu et al., 2021a) | 88.0 M | 15.4 G | 83.5 | 1.522 |
| MLP-Fusion-Swin-B (Wei et al., 2023) | 70.8 M | 13.3 G | 83.2 | 1.384 |
| NTK-SAP-Swin-B (Wang et al., 2023) | 72.6 M | 13.2 G | 83.2 | 1.369 |
| DeepCompress Swin-B (Ahmed et al., 2025) | 71.5 M | 13.0 G | 83.1 | 1.357 |
| **KCR-Swin-B (Ours)** | 70.2 M | 12.6 G | **84.7** | **1.312** |

to 70.2M, the FLOPs from $15.4$G to $12.6$G, and the inference time from $1.522$ ms/image to $1.312$ ms/image. These results underscore the practical deployment advantages of KCR regularization, which enhances accuracy and efficiency without sacrificing runtime performance.

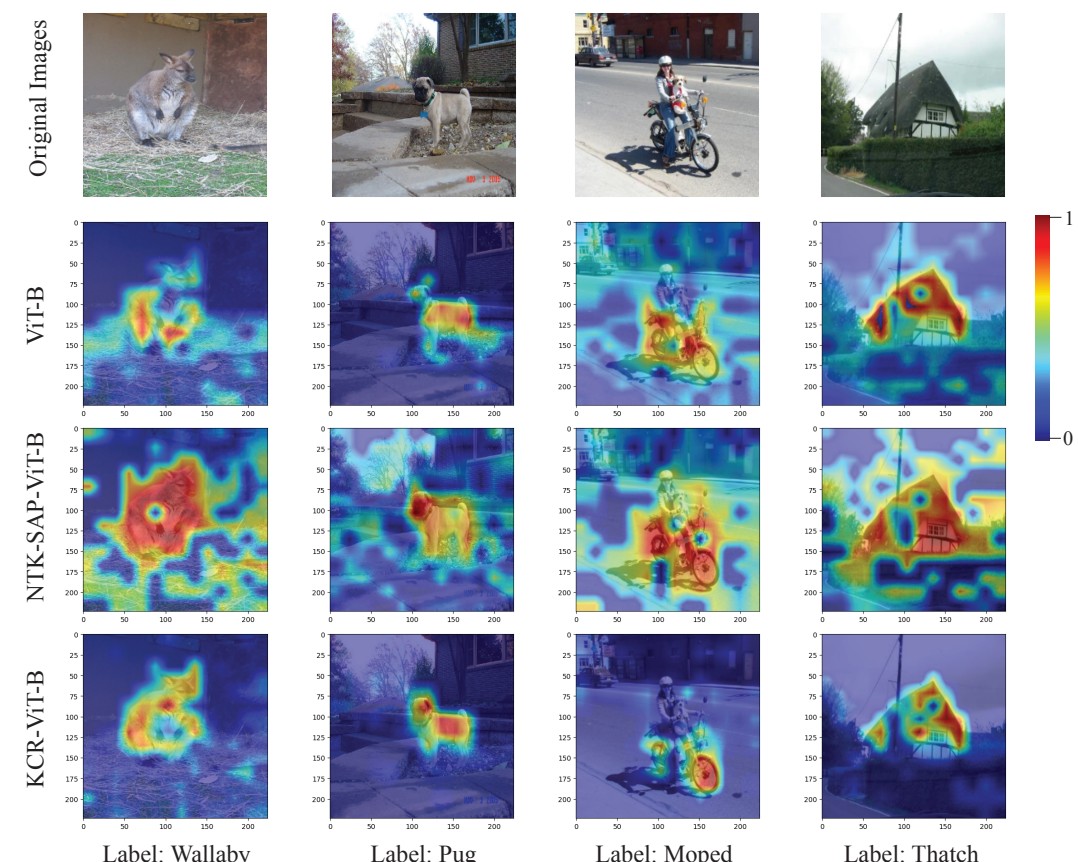

Figure 2: Grad-CAM visualization across four ImageNet classes, including wallaby, pug, moped, and thatch. The first row illustrates the original images. The second, third, and fourth row illustrates the Grad-CAM heatmaps from ViT-B (Dosovitskiy et al., 2021), NTK-SAP-ViT-B (Wang et al., 2023), and KCR-ViT-B, respectively.

## D.8 VISUALIZATION RESULTS

To qualitatively assess the discriminative capacity and spatial focus of different transformer variants, we apply the Grad-CAM technique (Selvaraju et al., 2017) to visualize the class-specific activation regions in the input images. We compute the Grad-CAM heatmaps for images from four representative ImageNet classes, including wallaby, pug, moped, and thatch, comparing the baseline ViT-B, NTK-SAP-ViT-B, and our KCR-ViT-B. Figure 2 illustrates that the proposed KCR-ViT-B consistently generates activation maps that are more spatially focused on the salient regions of the target object, while suppressing irrelevant background signals. In contrast, the attention maps of the ViT-B and NTK-SAP-ViT-B tend to exhibit higher activation in non-discriminative areas, such as sky, grass, or surrounding clutter, which may introduce unnecessary noise into the prediction process. The above observation suggests that the proposed KC regularization not only facilitates compression and computational efficiency but also enhances the model's representation learning capability for the classification task by guiding it to attend more selectively to task-relevant features.

## D.9 STUDY ON THE IMPACT OF KC

To assess the influence of KC on the performance of the KCR-Transformer, we perform an ablation study in which models have identical FLOPs and parameter sizes but different KC values. By varying the balancing weight $\eta$ of the KCR regularization term in the training loss, the resulting models achieve different KC levels while maintaining matched model sizes. It is observed in Table 15 that reducing KC up to a certain range improves classification accuracy, whereas overly aggressive KC reduction eventually leads to a slight degradation in performance.

Table 15: Imapct of KC on the Top-1 Accuracy. The study is performed on KCR-Swin-B. The first row in the table denotes the baseline uncompressed Swin-B model.

| $\eta$ | KC | Top-1 (%) |
|---|---|---|
| - | 3.21 | 83.5 |
| 0.01 | 1.75 | 83.9 |
| 0.05 | 0.87 | 84.3 |
| 0.2 | 0.52 | **84.7** |
| 0.4 | 0.52 | 84.6 |
| 0.6 | 0.49 | 84.5 |
| 0.8 | 0.47 | 84.4 |
| 1.0 | 0.45 | 84.5 |
| 1.5 | 0.43 | 84.5 |
| 2.0 | 0.41 | 84.4 |
| 2.5 | 0.40 | 84.3 |

## D.10 ABLATION STUDY ON THE KCR REGULARIZATION

To study the effect of the two-stage training procedure independently from the contribution of the kernel-based regularization, we performed an additional ablation study in which the KCR regularization is disabled by setting $\eta = 0$ while keeping the same two-stage training pipeline for channel selection. This baseline isolates the impact of the KCR term itself. It is observed in Table 16 that the two-stage training alone does not lead to an improvement in model performance. In contrast, incorporating the KCR regularization leads to consistent performance improvements across different architectures, which demonstrates that the improvement primarily arises from the kernel-based regularization, which is aligned with the theoretical analysis in Theorem 3.1. In particular, reducing the KC through the KCR regularization effectively lowers the upper bound for the generalization error, leading to better generalization capability and classification accuracy.

Table 16: Ablation Study on the Impact of KCR Regularization.

| Models | Top-1 (%) |
|---|---|
| ViT-B | 83.7 |
| KCR-ViT-B without KCR Regularization | 83.3 |
| KCR-ViT-B | **84.6** |
| Swin-B | 83.5 |
| KCR-Swin-B without KCR Regularization | 83.1 |
| KCR-Swin-B | **84.7** |

## D.11 VISUALIZATION OF CHANNEL SELECTION MASKS

To further improve the interpretability of the channel selection mechanism by the Gumbel-Softmax operation, we illustrate the learned channel-selection masks for KCR-ViT-S and KCR-ViT-B in Figure 3. Each row in the heatmap corresponds to a layer, and each column corresponds to a channel of that layer. As the Gumbel-Softmax temperature decrease in the search phase, the learned masks evolve toward near-binary channel-selection masks, with most probabilities pushed close to either

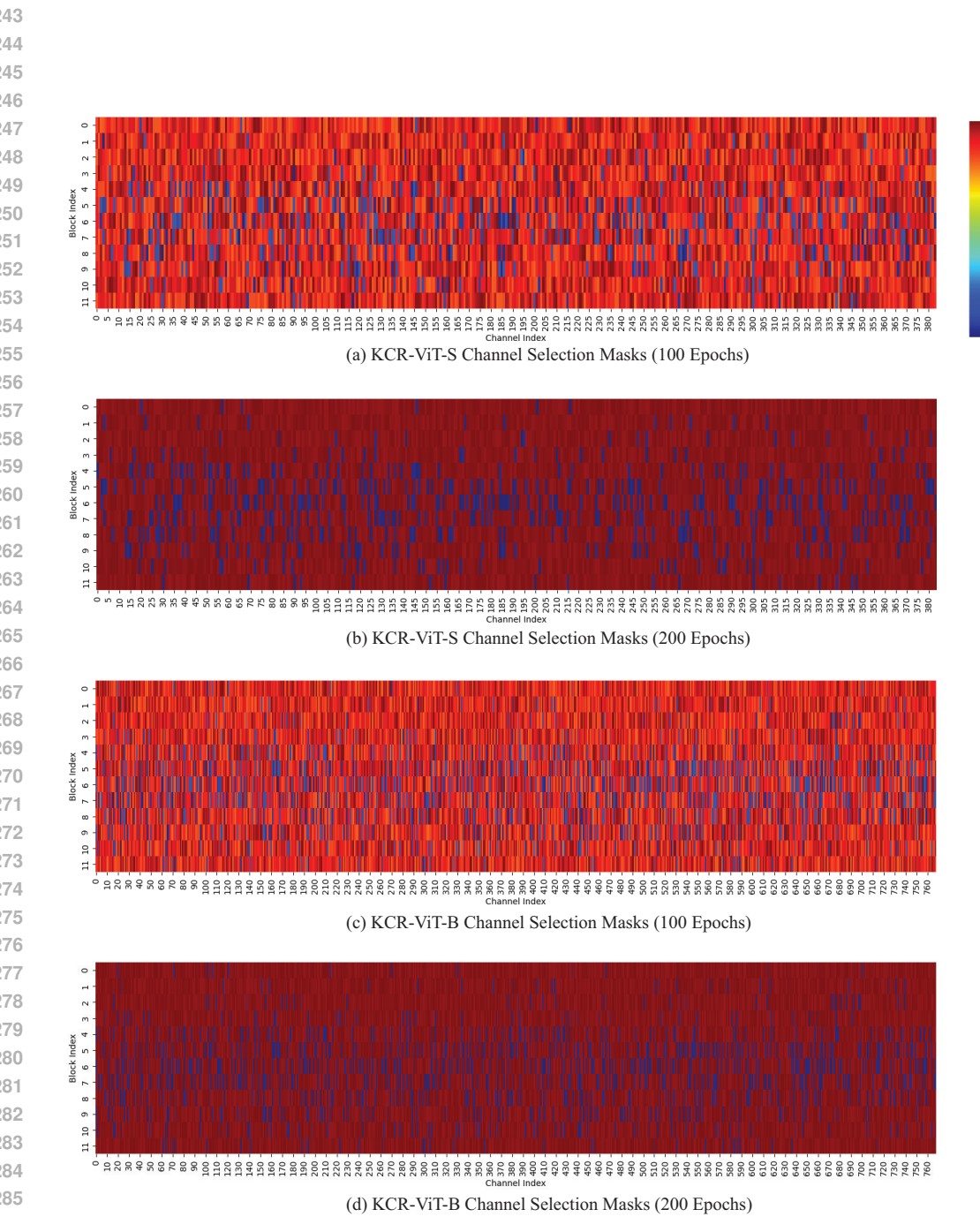

(a) KCR-ViT-S Channel Selection Masks (100 Epochs)

(b) KCR-ViT-S Channel Selection Masks (200 Epochs)

(c) KCR-ViT-B Channel Selection Masks (100 Epochs)

(d) KCR-ViT-B Channel Selection Masks (200 Epochs)

Figure 3: Visualization of learned Gumbel–Softmax channel-selection masks for KCR-ViT-S and KCR-ViT-B. Figures (a) and (b) illustrate the channel selection masks for ViT-S after 100 and 200 epochs in the search phase for KCR-ViT-S. Figures (c) and (d) illustrate the channel selection masks for ViT-S after 100 and 200 epochs in the search phase for KCR-ViT-B. The entire search phase takes 200 epcohs.

0 or 1 at the end of the search phase. Furthermore, we observe that the shallow layers retain a larger proportion of channels, whereas the middle layers undergo a more aggressive reduction in the number of channels preserved.

