# OpenReview forum: "Compression of Vision Transformer by Reduction of Kernel Complexity"
_ICLR.cc/2026/Conference — Submitted to ICLR 2026_

### Official Review · Reviewer_Ny7d · 2025-10-29

**Soundness:** 3
**Presentation:** 2
**Contribution:** 3
**Rating:** 6
**Confidence:** 4

**Summary:**

In this paper, a compression method for ViTs, namely KCR-Transformer,  is proposed. KCR-Transformer is a channel pruning method, which focuses on remove some of output channels of self-attentions to reduce the computation costs of MLP layers. To select the removeable channels, a decision mask g is defined, and according to the paper, g can be trained using gradient descent algorithm.  Furthermore, this paper includes theorietical proof to show that the KCR-Transformer can result in a tight upper and lower bound of the expected risk of DNN if the kernel complexity is small enough. This serves as a basis of the introducing of KCR blocks. Experimental results on variety of image-related tasks show that KCR-Transformers have lower computation costs but comparable or better performances comparing with their counterpart.

**Strengths:**

1. The proposed method find a suitable way to remove part of output channels of attention module. In this way, the computation costs of ViTs are reduced, but the network performances can be maintained.

2. This paper provides theorietical basis of the method, and shows that reducing KC is important for the final performance.

3. The experiments cover many image-related tasks, and the results can support the main arguements of the paper. Moreover, the ablation studies support theorem 3.1 well.

**Weaknesses:**

1. Some details of the method need to be included in the main paper. For instance:

(1) How to use gradient descent to train the decision mask g, and what the values of hyperparameters of g are used in each experiments.

(2) Why the decision mask g should multiply with both input and output features of MLP layers?  What is the difference between this method and multiplying the pruned input of MLP with a smaller weight matrix to get a smaller output?

(3) Some details, such as experiment configurations, should be move into the main paper.

**Questions:**

My questions are included in the part of Weaknesses.

---

> ### Author Response · Authors · 2025-11-26
> **Response to Official Review by Reviewer Ny7d**
>
> We appreciate the review and the suggestions in this review. The raised issues are addressed below.
>
> **Responses to the Weaknesses**
>
> **1. "How to use gradient descent to train the decision mask g, and what the values of hyperparameters of g are used in each experiments."**
>
>
> The decision mask $g$ is trained via standard gradient descent by optimizing its corresponding sampling parameters $\alpha$ through the Gumbel Softmax relaxation, which enables differentiable channel selection. As detailed in Section C of the appendix, the hyperparameters associated with $g$ follow common settings in differentiable pruning and NAS literature [1,2,3]. The temperature $\tau$ is initialized at 4.5 and decayed by a factor of 0.95 every epoch over the search phase, and the Gumbel noise is sampled per mini-batch from a standard Gumbel(0, 1) distribution.
>
> **2. "Why the decision mask g should multiply with both input and output features of MLP layers?..."**
>
> The decision mask is applied to both the input and output of the MLP layers because, in standard transformer blocks, the MLP input is exactly the output of the multi-head self-attention module, and both share the same channel dimension due to the residual connection [4, 5, 6]. Consequently, pruning the MLP input is equivalent to pruning the channels of the attention output. As discussed in Section 3.1, multiplying the decision mask with both the MLP input and output ensures that the pruned representation stays in a consistent reduced channel space. Such a design follows established practice in Vision Transformer channel-pruning literature [4, 5, 6], where both input and output channels of the MLP layers are pruned jointly to maintain architectural consistency.
>
> **3. "Some details, such as experiment configurations, should be move into the main paper."**
>
> We have moved the key experiment configurations from Section C of the appendix of the original paper to Section 4.1 of the main paper of the revised paper.
>
> **References**
>
> [1] Wan, Alvin, et al. "Fbnetv2: Differentiable neural architecture search for spatial and channel dimensions." Proceedings of the IEEE/CVF conference on computer vision and pattern recognition. 2020.
>
> [2] Chang, Jianlong, et al. "Data: Differentiable architecture approximation." Advances in Neural Information Processing Systems 32 (2019).
>
> [3] Wu, Bichen, et al. "Fbnet: Hardware-aware efficient convnet design via differentiable neural architecture search." Proceedings of the IEEE/CVF conference on computer vision and pattern recognition. 2019.
>
> [4] Chen, Tianlong, et al. "Chasing sparsity in vision transformers: An end-to-end exploration." Advances in Neural Information Processing Systems 34 (2021).
>
> [5] Chavan, Arnav, et al. "Vision transformer slimming: Multi-dimension searching in continuous optimization space." Proceedings of the IEEE/CVF Conference on Computer Vision and Pattern Recognition. 2022.
>
> [6] Zheng, Li, et al. "Pruning vision transformers using structural reparameterization." Advances in Neural Information Processing Systems 35 (2022).

---

### Official Review · Reviewer_SStQ · 2025-10-31

**Soundness:** 2
**Presentation:** 2
**Contribution:** 1
**Rating:** 2
**Confidence:** 4

**Summary:**

The paper demonstrates their research on transformer architecture design that both benefits from channel selection in the MLP layers for attention outputs with Gumbel Softmax and generalization-aware pruning with the proposed KC metric. The authors report that their model achieves improved performance on classification, object detection, semantic segmentation, and VQA tasks, all while reducing computational overhead.

**Strengths:**

1.	The authors provide a solid and detailed proof for their main theory, Theorem 3.1, and comprehensively used the theorem guiding them to train their model.
2.	Excessive experiments on 4 different types of tasks are conducted. Promising results in all fields are reported.

**Weaknesses:**

1.	The practical benefit of the theorem 3.1 and the according adapted KCR loss function (2) using KC lacks further evidence. The contribution of including the KC as regularization term is not proven in the ablation study in section 4.3. The section merely provides the effectiveness of approximate TNN on regularizing the KC metric but not explicitly improving the accuracy. Experimental results without KCR term might help in explaining this.
2.	The claim of generalization-awareness in Section 3.2 is purely theoretical and lacks empirical backing, for instance, providing zero-shot experiments.
3.	The paper’s explainability would be significantly improved with more figures, such as heat maps of the Gumbel-Softmax channel selection or figures that explicitly visualize the generalization performance.
4.	The contributions appear to be incremental, as the work builds on existing ones on Gumbel-Softmax channel selection and neural architecture searching techniques. The paper's main theoretical contribution (KCR) lacks the necessary empirical validation to prove its effectiveness.

**Questions:**

1.	How does KCR help in terms of accuracy, efficiency and generalization? Could the authors provide more direct experimental evidence (e.g., targeted ablations)?
2.	How exactly is the KCR involved in the whole model and training process? (Only the loss function part according to my understanding)

---

> ### Author Response · Authors · 2025-11-26
> **Response to Official Review by Reviewer SStQ (Part 1)**
>
> We appreciate the review and the suggestions in this review. The raised issues are addressed below.
>
> **Responses to the Weaknesses**
>
> **1. "The practical benefit of the theorem 3.1 and the according adapted KCR loss function (2) using KC lacks further evidence..."**
>
> To study the impact of the KCR regularization, we performed an additional ablation study in which the KCR regularization is disabled by setting $\eta = 0$ while keeping the same two-stage training pipeline for channel selection. This baseline isolates the impact of the KCR regularization itself. It is observed that the two-stage training alone does not lead to an improvement in model performance. In contrast, incorporating the KCR regularization leads to consistent performance improvements across different architectures, which demonstrates that the improvement primarily arises from the kernel-based regularization, which is aligned with the theoretical analysis in Theorem 3.1. In particular, reducing the kernel complexity (KC) through the KCR regularization effectively reduces the upper bound for the generalization error, leading to better generalization capability and classification accuracy.
>
> The results are also added to Table 16 in Section D.10 of the revised paper, which are also shown below.
>
> Table: Ablation Study on the Impact of KCR Regularization
> | Models    | Top-1 (%) |
> |-------|----|
> | ViT-B         | 83.7      |
> | KCR-ViT-B without KCR Regularization        | 83.3      |
> | **KCR-ViT-B**      | **84.6**  |
> | Swin-B           | 83.5      |
> | KCR-Swin-B without KCR Regularization       | 83.1      |
> | **KCR-Swin-B**    | **84.7**  |
>
> Furthermore, to assess the influence of kernel complexity (KC) on the performance of the KCR-Transformer, we conducted an ablation study in which models have identical FLOPs and parameter sizes but different KC values. By varying the balancing weight $\eta$ of the KCR regularization term in the training loss, the resulting models achieve different KC levels while maintaining matched model sizes. It is observed in the table below that reducing KC up to a certain range improves classification accuracy, whereas overly aggressive KC reduction eventually leads to a slight degradation in performance. The results are shown below and also in Table 15 in Section D.9 of the revised paper.
>
> Table: Impact of KC on Top-1 Accuracy (KCR-Swin-B).  The first row corresponds to the baseline Swin-B model.
> | $\eta$    | KC    | Top-1 (%) |
> |------|-------|-----------|
> | -    | 3.21  | 83.5      |
> | 0.01 | 1.75  | 83.9      |
> | 0.05 | 0.87  | 84.3      |
> | 0.2  | 0.52  | **84.7**  |
> | 0.4  | 0.52  | 84.6      |
> | 0.6  | 0.49  | 84.5      |
> | 0.8  | 0.47  | 84.4      |
> | 1.0  | 0.45  | 84.5      |
> | 1.5  | 0.43  | 84.5      |
> | 2.0  | 0.41  | 84.4      |
> | 2.5  | 0.40  | 84.3      |
>
> **2. "The claim of generalization-awareness in Section 3.2 is purely theoretical and lacks empirical backing, for instance, providing zero-shot experiments."**
>
> We emphasize that the generalization capability of DNNs in this paper refers to the DNNs’ prediction performance on the test data, and a stronger generalization capability corresponds to higher prediction accuracy (top-1 accuracy) on the test data [1]. KCR regularization improves the generalization capability of DNNs by lowering the upper bound for the generalization error defined in Eq. (1), which is the KCR upper bound in Theorem 3.1. The effectiveness of generalization-aware compression by KCR regularization is shown in our response to weakness 1 and Table 1 of the revised paper for the image classification task and Tables 2,3,5,6, and 7 of the revised paper for the downstream tasks.
>
> **3. "The paper’s explainability would be significantly improved with more figures, such as heat maps of the Gumbel-Softmax channel selection ..."**
>
> To further improve the interpretability of the channel selection mechanism by the Gumbel-Softmax operation, we illustrate the learned channel-selection masks for KCR-Transformers in Figure 3 in Section D.11 of the revised paper. Each row in the heatmap corresponds to a layer, and each column corresponds to a channel of that layer.

---

> > ### Author Response · Authors · 2025-11-26
> > **Response to Official Review by Reviewer SStQ (Part 2)**
> >
> > **4. "The contributions appear to be incremental, as the work builds on existing ones on Gumbel-Softmax channel selection..."**
> >
> > **We respectfully point out it is a significant misunderstanding that “The contributions appear to be incremental…The paper's main theoretical contribution (KCR) lacks the necessary empirical validation to prove its effectiveness. ”. The KCR regularization term, theoretically motivated by our theoretical contribution in Theorem 3.1, is in fact added to the training loss in Eq. (2) when performing both searching and retraining stages of generalization-aware compression. The effectiveness of generalization-aware compression by KCR regularization is shown in Table 1 of the revised paper for the image classification task and Tables 2,3,5,6, and 7 of the revised paper for the downstream tasks, where all the results are obtained by generalization-aware compression by KCR regularization. As a result, it is a factual mistake to claim that “KCR lacks the necessary empirical validation to prove its effectiveness”.**
> >
> > To strengthen the empirical validation of the impact of KCR on the performance of the KCR-Transformer, we conducted an ablation study in which models have identical FLOPs and parameter sizes but different KC values. By varying the balancing weight $\eta$ of the KCR regularization term in the training loss, the resulting models achieve different KC levels while maintaining matched model sizes. It is observed in the table below that reducing KC consistently improves the classification accuracy of the KCR-Transformers over the baseline models. The ablation study results are also added to Table 15 in Section D.9 of the revised paper.
> >
> >
> >
> > **Responses to the Questions**
> >
> > **1. "How does KCR help in terms of accuracy, efficiency and generalization?..."**
> >
> > The KCR regularization reduces the Kernel Complexity (KC) of the KCR-Transformer, which lowers the upper bound for the generalization error, leading to better generalization capability and classification accuracy.
> > Please refer to our response to Weakness 4, where we have performed a targeted ablation study showing the impact of the value of KC on the classification accuracy of the KCR-Transformer. The training and searching cost of the KCR-Transformer is studied in Table 9 in Section D.5 of the revised paper.
> > It is observed that KCR consistently improves the generalization capability of the KCR-Transformers, leading to higher top-1 accuracy while introducing only marginal additional training cost.
> > Moreover, we emphasize here again that the generalization capability in this paper refers to the prediction accuracy of the model on the unseen test data.
> >
> > **2. "How exactly is the KCR involved in the whole model and training process? (Only the loss function part according to my understanding)"**
> >
> > The KCR regularization term is in both the search loss, $\mathcal L_{\textup{search},j}^{(t)} (\mathcal W,\alpha)$ described in Section 3.3, and the retraining loss $\mathcal L_{\textup{train,j}}^{(t)}(\mathcal W)$ in Eq. (2). This is the way KCR is involved our generalization-aware compression process.
> >
> > **References**
> >
> > [1] Bishop, Christopher M., and Nasser M. Nasrabadi. Pattern recognition and machine learning. Vol. 4. No. 4. New York: springer, 2006.

---

### Official Review · Reviewer_HaUA · 2025-11-01

**Soundness:** 3
**Presentation:** 2
**Contribution:** 3
**Rating:** 6
**Confidence:** 3

**Summary:**

This paper proposes a two-stage training framework for network pruning, aiming to balance model efficiency and performance. In the Search Stage, the weights are optimized with cross-entropy loss and Kernel Regularization, while channel gates are updated with an additional computation cost constraint, and the gate values are set to 0/1 via temperature annealing to finalize the pruned structure. Then, in the Retraining Stage, the fixed pruned structure is fine-tuned to achieve efficient model compression while preserving original model’s performance.

**Strengths:**

- The overall pipeline is clearly presented with a detailed formula and explanation.
- The method is well motivated, followed by a two-stage design to help find the efficient structure of the model and recover the model performance with fine-tuning.
- Experiments across different backbones consistently show better performance compared to the baseline method and other model compression methods, with fewer parameters and FLOPS.

**Weaknesses:**

- For Theorem 3.1 and its proof, it is unclear to me what $r^*$ represents. (Lines 859 and 860)
- In section 3.2, Line 226 states that $F \in R^{n\times d}$, but line 228 further claims $K = F^T F \in R^{n\times n}$. Is there a contradiction here?
- It would be better for the authors to add an ablation study to verify whether the observed improvement is attributed to the designed two-stage training method or the kernel function acting as a regularizer.

**Questions:**

- It is common practice in network pruning to employ a fixed decision mask for sample selection, yet the paper adopts a less conventional approach: optimizing the sampling parameter $\alpha$ via gradient descent. This choice lacks explicit justification, for instance, why is gradient-based optimization more effective, as it may introduce training instability or overfitting.
- Algorithm 1 specifies that only 30% of samples are used to update $\alpha$ (with 70% for weight updates), but the rationale behind this 30% ratio remains unclear. It would be valuable to conduct an ablation study to verify how varying this sample split ratio impacts $\alpha$’s ability to select meaningful channels, as well as to further explain why this specific ratio was chosen over alternatives (e.g., 20%, 40%) or adaptive splitting strategies.

---

> ### Author Response · Authors · 2025-11-26
> **Response to Official Review by Reviewer HaUA (Part 1)**
>
> We appreciate the review and the suggestions in this review. The raised issues are addressed below.
>
> **Responses to the Weaknesses**
>
> **1. "For Theorem 3.1 and its proof, it is unclear to me what $r^*$ represents. (Lines 859 and 860)."**
>
> Response: $r^*$ is the fixed point of the kernel complexity of the dynamic kernel $K$, and the dynamic kernel $K$ is defined in line 221 of the revised paper. Please refer to  Eq. (8) in [6] for the details about the fixed point of the kernel complexity, which is also referred to as the critical population rate in [6].
>
> **2. "In section 3.2, Line 226 states that $F\in R^{n \times d}$, but line 228 further claims $K=F^{\top}F\in R^{n \times n}$. Is there a contradiction here?"**
>
> This is a typo which is fixed in the revised paper, and $K=F F^{\top}\in R^{n \times n}$.
>
> **3. "It would be better for the authors to add an ablation study to verify whether the observed improvement is attributed to the designed two-stage training method or the kernel function acting as a regularizer."**
>
> To study the effect of the two-stage training procedure independently from the contribution of the kernel-based regularization, we performed an additional ablation study in which the KCR regularization is disabled by setting $\eta = 0$ while keeping the same two-stage training pipeline for channel selection. This baseline isolates the impact of the KCR term itself. It is observed that the two-stage training alone does not lead to an improvement in model performance. In contrast, incorporating the KCR regularization leads to consistent performance improvements across different architectures, which demonstrates that the improvement primarily arises from the kernel-based regularization, which is aligned with the theoretical analysis in Theorem 3.1. In particular, reducing the kernel complexity (KC) through the KCR regularization effectively lowers the upper bound for the generalization error, leading to better generalization capability and classification accuracy.
> The results are also added to Table 16 in Section D.10 of the revised paper, which are also shown below.
>
> Table: Ablation Study on the Impact of KCR Regularization
> | Models        | Top-1 (%) |
> |---|---|
> | ViT-B         | 83.7      |
> | KCR-ViT-B without KCR Regularization        | 83.3      |
> | **KCR-ViT-B**      | **84.6**  |
> | Swin-B           | 83.5      |
> | KCR-Swin-B without KCR Regularization       | 83.1      |
> | **KCR-Swin-B**                              | **84.7**  |
>
>
> Furthermore, to assess the influence of kernel complexity (KC) on the performance of the KCR-Transformer, we conducted an ablation study in which models have identical FLOPs and parameter sizes but different KC values. By varying the balancing weight $\eta$ of the KCR regularization term in the training loss, the resulting models achieve different KC levels while maintaining matched model sizes. It is observed in the table below that reducing KC up to a certain range improves classification accuracy, whereas overly aggressive KC reduction eventually leads to a slight degradation in performance. The results are shown below and also in Table 15 in Section D.9 of the revised paper.
>
> Table: Impact of KC on Top-1 Accuracy (KCR-Swin-B).  The first row corresponds to the baseline Swin-B model.
> | $\eta$    | KC    | Top-1 (%) |
> |------|-------|-----------|
> | -    | 3.21  | 83.5      |
> | 0.01 | 1.75  | 83.9      |
> | 0.05 | 0.87  | 84.3      |
> | 0.2  | 0.52  | **84.7**  |
> | 0.4  | 0.52  | 84.6      |
> | 0.6  | 0.49  | 84.5      |
> | 0.8  | 0.47  | 84.4      |
> | 1.0  | 0.45  | 84.5      |
> | 1.5  | 0.43  | 84.5      |
> | 2.0  | 0.41  | 84.4      |
> | 2.5  | 0.40  | 84.3      |
>
> **4. "It is common practice in network pruning to employ a fixed decision mask for sample selection, yet the paper adopts a less conventional approach: optimizing the sampling parameter via gradient descent..."**
>
> The sampling parameter $\alpha$ in Gumbel-Softmax is optimized by gradient descent, which decides the decision mask, and is a standard and widely adopted practice in differentiable architecture search and differentiable pruning methods, such as Gumbel-Softmax based channel selection [1, 2] and differentiable Neural Architecture Search (NAS) [3,4, 5]. This approach enables end-to-end optimization of both the architecture parameters and the network weights within a unified training objective, which has been shown to be stable and effective in numerous prior works [1, 2, 3, 4, 5]. In our setting, $\alpha$ is the architecture parameter optimized jointly with the model parameters and does not introduce training instability. Instead, such a differentiable design allows the channel-selection decisions to be guided directly by the training loss and the KCR regularization.

---

> > ### Author Response · Authors · 2025-11-26
> > **Response to Official Review by Reviewer HaUA (Part 2)**
> >
> > **5. "Algorithm 1 specifies that only 30% of samples are used to update $\alpha$ (with 70% for weight updates), but the rationale behind this 30% ratio remains unclear..."**
> >
> > We follow existing works [3, 5] in using only 30% of samples to update the architecture parameters $\alpha$, and the remaining 70% samples are used to update the network weights. Let $\phi$ denote the ratio of the data used for updating the architecture parameters. To study the impact of $\phi$ on the performance of the searched KCR models. We conducted an ablation study by varying the ratio $\phi$ from 10% to 50% with a step size of 10%. It is observed in the table below that the performance of the searched architecture is not sensitive to the value of $\phi$.
> >
> > Table: Impact of the ratio of the data used for updating the architecture parameters ($\phi$) on KCR-ViT-B Performance
> > | $\phi$ | Top-1 (%) |
> > |-----|-----------|
> > | 0.10       | 84.5      |
> > | 0.20    | 84.6      |
> > | 0.30  | 84.6  |
> > | 0.40   | 84.4      |
> > | 0.50    | 84.6      |
> >
> >
> > **References**
> >
> > [1] Kang, Minsoo, and Bohyung Han. "Operation-aware soft channel pruning using differentiable masks." International conference on machine learning. PMLR, 2020.
> >
> > [2] Herrmann, Charles, Richard Strong Bowen, and Ramin Zabih. "Channel selection using gumbel softmax." European conference on computer vision. Cham: Springer International Publishing, 2020.
> >
> > [3] Wan, Alvin, et al. "Fbnetv2: Differentiable neural architecture search for spatial and channel dimensions." Proceedings of the IEEE/CVF conference on computer vision and pattern recognition. 2020.
> >
> > [4] Chang, Jianlong, et al. "Data: Differentiable architecture approximation." Advances in Neural Information Processing Systems 32 (2019).
> >
> > [5] Wu, Bichen, et al. "Fbnet: Hardware-aware efficient convnet design via differentiable neural architecture search." Proceedings of the IEEE/CVF conference on computer vision and pattern recognition. 2019.
> >
> > [6] Raskutti et al. Early Stopping and Non-parametric Regression: An Optimal Data-dependent Stopping Rule. JMLR 2014.

---

### Official Review · Reviewer_LddC · 2025-11-01

**Soundness:** 3
**Presentation:** 2
**Contribution:** 2
**Rating:** 4
**Confidence:** 4

**Summary:**

This paper introduces the KCR-Transformer, a method that compresses Vision Transformers by using a Kernel Complexity (KC) generalization bound to guide the differentiable channel pruning of its MLP layers.

**Strengths:**

KCR-Transformer is shown to effectively reduce FLOPs, parameters, and the KC metric, while achieving strong empirical results that often surpass the original, uncompressed models in accuracy.

**Weaknesses:**

Unclear Training Cost and Scalability:
The proposed KCR-Transformer introduces several additional computational components, including Gumbel-Softmax-based channel selection, the TNN regularization term, and a complex two-stage training process (search + retraining). These modules inevitably increase the total training-time computation and memory usage compared to baseline models. While the paper provides a Minutes/Epoch comparison for the retraining phase in Appendix D.5 (Table 9), this analysis is incomplete. It fails to quantify the total training cost, most notably the significant overhead from the entire search phase.

Lack of Fine-Grained Ablation on the KC-Performance:
The critical ablation is absent: comparing models of identical compressed size while demonstrating performance differences solely by varying the corresponding KC values is necessary, otherwise the claim that KC drives performance remains unproven.

**Questions:**

See weakness

---

> ### Author Response · Authors · 2025-11-26
> **Response to Official Review by Reviewer LddC**
>
> We appreciate the review and the suggestions in this review. The raised issues are addressed below.
>
> **Responses to the Weaknesses**
>
> **1. "...While the paper provides a Minutes/Epoch comparison for the retraining phase in Appendix D.5 (Table 9), this analysis is incomplete. It fails to quantify the total training cost, most notably the significant overhead from the entire search phase..."**
>
> To provide a complete picture of the training cost, we have added the time for the search process and the retraining process to Table 9 in Section D.5 of the revised paper, which reports the total end-to-end training time of KCR-Transformer, including both the search stage and the retraining stage.
> The table is also attached below. It is observed that the search phase introduces only a moderate overhead relative to the full training budget, which costs less than 7.2% of the baseline model’s training time. Moreover, KCR achieves a substantial improvement in top-1 classification accuracy over the corresponding baseline models.
>
> Table: Training time evaluation on the training set of ImageNet-1K. The comparison is performed on one NVIDIA A100 40GB GPU with full precision.
> | Model   | # Params | FLOPs | Top-1  | Search Time (Hours) | Re-Training Time (Hours) |
> |--|--|--|--|--|--|
> | ViT-S     | 22.1 M  | 4.3 G | 81.2  | -   | 84.0         |
> | KCR-ViT-S  | 19.8 M  | 3.8 G | 82.2 | 6.03     | 88.5      |
> | ViT-B | 86.5 M  | 17.6 G | 83.7   | -   | 167.5     |
> | KCR-ViT-B | 69.5 M  | 14.5 G | 84.6 | 11.7  | 173.4   |
> | Swin-T | 29.0 M  | 4.5 G | 81.3   | -  | 104.0  |
> | KCR-Swin-T | 24.6 M | 3.9 G | 82.4 | 7.3| 108.5 |
> | Swin-B | 88.0 M  | 15.4 G | 83.5  | - | 191.5 |
> | KCR-Swin-B    | 70.2 M  | 12.6 G | 84.7 | 13.3 | 197.1|
>
>
> **2."Lack of Fine-Grained Ablation on the KC-Performance..."**
>
> To assess the influence of kernel complexity (KC) on the performance of the KCR-Transformer, we conducted an ablation study in which models have identical FLOPs and parameter sizes but different KC values. By varying the balancing weight $\eta$ of the KCR regularization term in the training loss, the resulting models achieve different KC levels while maintaining matched model sizes. It is observed in the table below that reducing KC up to a certain range improves classification accuracy, whereas overly aggressive KC reduction eventually leads to a slight degradation in performance.
>
> Table: Impact of KC on Top-1 Accuracy (KCR-Swin-B).  The first row corresponds to the baseline Swin-B model.
> | $\eta$ | KC| Top-1 (%) |
> |--|--|--|
> | -    | 3.21  | 83.5  |
> | 0.01 | 1.75  | 83.9  |
> | 0.05 | 0.87  | 84.3  |
> | 0.2  | 0.52  | 84.7 |
> | 0.4  | 0.52  | 84.6  |
> | 0.6  | 0.49  | 84.5  |
> | 0.8  | 0.47  | 84.4  |
> | 1.0  | 0.45  | 84.5  |
> | 1.5  | 0.43  | 84.5  |
> | 2.0  | 0.41  | 84.4  |
> | 2.5  | 0.40  | 84.3  |
>
> The results are also added to Table 15 in Section D.9 of the revised paper.

---

### Author Response · Authors · 2025-12-02
**Summary of Revisions (Part 1)**

Dear AC,

Thank you for your time handling and reviewing this paper. We have thoroughly addressed all reviewers’ concerns, which is also reflected in the revised paper. Below is a concise summary of our key clarifications and revisions addressing all the concerns.

 (1) First, **there is a significant factual misunderstanding in the comment of Reviewer SStQ that “the contributions appear incremental and lack empirical validation”, and “The contributions appear to be incremental…The paper's main theoretical contribution (KCR) lacks the necessary empirical validation to prove its effectiveness. ”**. The KCR regularization term, theoretically motivated by our theoretical contribution in Theorem 3.1, is empirically justified in all the experimental results of this paper. In fact, the KCR regularization term is theoretically motivated by the fact that the reduction of the Kernel Complexity (KC) reduces the generalization error bound of DNNs, and the KCR regularization term is in fact the critical part of the training loss in Eq. (2) when performing both searching and retraining stages of generalization-aware compression for all the experiments in this paper. The effectiveness of generalization-aware compression by KCR regularization is shown in Table 1 of the revised paper for the image classification task and Tables 2,3,5,6, and 7 of the revised paper for the downstream tasks, where all the results are obtained by generalization-aware compression by KCR regularization. As a result, it is a factual mistake to claim that “KCR lacks the necessary empirical validation to prove its effectiveness”.

(2) **Reviewer SStQ's comment that  "The claim of generalization-awareness... lacks empirical backing, for instance, providing zero-shot experiments" has another significant factual misunderstanding.**
**"Generalization-awareness" in this paper refers specifically to test-set generalization (i.e., how well the DNN predicts on unseen data from the same distribution). It does NOT refer to cross-task generalization such as zero-shot learning.** We emphasize that the generalization capability of DNNs in this paper refers to the DNNs’ prediction performance on the test data, and a stronger generalization capability corresponds to higher prediction accuracy (top-1 accuracy) on the test data (please refer to reference [1] in our rebuttal). KCR regularization improves the generalization capability of DNNs by lowering the upper bound for the generalization error defined in Eq. (1), which is the KCR upper bound in Theorem 3.1. The effectiveness of generalization-aware compression by KCR regularization is shown in our response to weakness 1 and Table 1 of the revised paper for the image classification task and Tables 2,3,5,6, and 7 of the revised paper for the downstream tasks.

(3) Reviewer LddC and Reviewer SStQ are concerned about the lack of an ablation study showing the influence of KC on the performance of the KCR-Transformer. To address this concern, we have added the required ablation study showing the relation between Kernel Complexity (KC) and its influence on the generalization of DNNs. A new fine-grained ablation study (Table 15, Section D.9) shows the relation between KC and generalization of DNNs by varying KC through different balancing weight $\eta$. The results show that different levels of KC reduction always improve the top-1 classification accuracy of compressed models, and overly aggressive KC reductions still render compressed models outperforming the uncompressed baseline. This trend matches our theoretical analysis: reducing KC tightens the upper bound on the generalization error in Theorem 3.1, leading to improved generalization capability. Furthermore, we added a targeted ablation (Table 16, Section D.10) where the KCR regularization is disabled ( $\eta$ = 0) while retaining the same two-stage compression pipeline (searching and retraining). It is observed that the two-stage training alone does not lead to an improvement in model performance. In contrast, incorporating the KCR regularization leads to consistent performance improvements across different architectures, which demonstrates that the improvement primarily arises from the KCR regularization, which is aligned with the theoretical analysis in Theorem 3.1.

(4) Reviewer LddC is concerned about the total training cost of the KCR-Transformer. To address this concern. the revised Table 9 in Section D.5 now reports the full end-to-end training cost of the KCR-Transformer, including both the search process and the retraining stage. The results show that the search phase introduces only a marginal overhead, which is less than 7.2% of the baseline model’s full training time, while consistently improving top-1 accuracy. This resolves the reviewer’s concern that the original comparison was incomplete and demonstrates that KCR delivers measurable benefits at modest cost.

---

> ### Author Response · Authors · 2025-12-02
> **Summary of Revisions (Part 2)**
>
> (5) Reviewer SStQ is concerned that the claim of generalization-awareness in Section 3.2 is purely theoretical and lacks empirical backing, and asked how KCR helps in terms of accuracy, efficiency, and generalization.
> To address this concern, we first emphasize that the generalization capability of DNNs in this paper refers to the DNNs’ prediction performance on the test data, and a stronger generalization capability corresponds to higher prediction accuracy (top-1 accuracy) on the test data. KCR regularization improves the generalization capability of DNNs by lowering the upper bound for the generalization error defined in Eq. (1), which is the KCR upper bound in Theorem 3.1. The effectiveness of generalization-aware compression by KCR regularization is shown in Table 1 of the revised paper for the image classification task and Tables 2,3,5,6, and 7 of the revised paper for the downstream tasks. The training and searching cost of the KCR-Transformer is studied in Table 9 in Section D.5 of the revised paper. It is observed that KCR consistently improves the generalization capability of the KCR-Transformers, leading to higher top-1 accuracy while introducing only marginal additional training cost.
>
> (6) Reviewer SStQ is concerned about the explainability of the channel selection masks.
> To address this concern, we have enhanced the interpretability of the method by adding heatmaps of the learned Gumbel-Softmax channel selection masks in Figure 3 in Section D.11 of the revised paper. These visualizations show sharp, near-binary selection patterns and clear layer-dependent pruning behavior, providing intuitive insight into the channel selection mechanism induced by our KCR regularization. We further clarified that the KCR regularization term appears in both the search loss and the retraining loss, ensuring that KCR influences the entire generalization-aware compression process.
>
> We have also addressed the issues from Reviewer HaUA and Reviewer Ny7d in our rebuttals and the revised paper.

---

### Meta-Review · Area_Chair_qs79 · 2026-01-07

**Summary:**

The paper proposed ViT compression method based on reducing Kernel-Complexity (KC). The paper first provides detailed theoretical evidence on the impact of reduced KC on the generalization performance. The paper proposes a loss function, KCR, to train the network while reducing the KC, which is connected to network compression.

The rebuttal resolves concerns about missing details and ablation studies. But, still, the concerns related to practical impacts and empirical evidence for generalization performance remain unresolved. Overall, the reviewers present mixed opinions on the paper.

To make a solid decision, I also review the paper and find significant problems with the experiment. Most of all, the comparison doesn't look valid. In Table 1-5, the proposed KCR is compared with three compression methods: MLP-Fusion, NTK-SAP, and DeepCompress. These are recent compressions with powerful performance. However, in Table 1-5, three methods show similar performance with the baseline or even underperform it. Intuitively, it is not a valid result considering their performance demonstrated in the original literature.
Additionally, it is necessary to report actual speed, such as throughput and latency, to demonstrate the contribution as a practical compression method.

I believe the reliability of the comparison is insufficient in all tables. It doesn't prove that KCR outperforms the existing method. There might be a reason, such as `NTK and compression methods are not effective on ViT`. However, if such an issue exists, it should be noted and discussed in the paper. Therefore, my decision leans toward rejection.

Overall, the reviewers present mixed opinions. I found significant concerns in the experiment. So, I can't recommend acceptance of the paper.

**Reviewer Concerns:**

Resolved concern
- Reviewer LddC
  - Training cost
  - Lack of fine-grained ablation
- Reviewer HaUA
  - Detailed explanation and error in the formula
  - Ablation study for two-stage training and kernel function acting
- Reviewer SStQ
  - Ablation study for KCR regularization
- Reviewer Ny7d
  - Missing details

Remaining concern
- Reviewer SStQ
  - Practical evidence of KCR is insufficient
  - No empirical evidence for generalization-awareness
  - Contribution looks incremental

**Reviewer Scores:**

- Reviewer LddC: Would raise the score.
- Reviewer HaUA: Would maintain the current score.
- Reviewer SStQ: Would maintain the current score.
- Reviewer Ny7d: Would maintain the current score.

---

### Decision · Program_Chairs · 2026-01-26

Reject